# SIMPL: Scalable and hassle-free optimisation of neural representations from behaviour

**Tom M George**
Sainsbury Wellcome Centre, UCL
`tom.george.20@ucl.ac.uk`

**Pierre Glaser**
Gatsby Computational Neuroscience Unit, UCL

**Kimberly Stachenfeld**
Google DeepMind & Columbia University

**Caswell Barry**
Dept. of Cell and Developmental Biology, UCL

**Claudia Clopath**
Sainsbury Wellcome Centre, UCL & Imperial College London
`c.clopath@imperial.ac.uk`

## Abstract

Neural activity in the brain is known to encode low-dimensional, time-evolving, behaviour-related variables. A long-standing goal of neural data analysis has been to identify these variables and their mapping to neural activity. A productive and canonical approach has been to simply visualise neural "tuning curves" as a function of behaviour. In reality, significant discrepancies between behaviour and the true latent variables—such as an agent thinking of position Y whilst located at position X—distort and blur the tuning curves, decreasing their interpretability. To address this, latent variable models propose to learn the latent variable from data; these are typically expensive, hard to tune, or scale poorly, complicating their adoption. Here we propose SIMPL (Scalable Iterative Maximization of Population-coded Latents), an EM-style algorithm which iteratively optimises latent variables and tuning curves. SIMPL is fast, scalable and exploits behaviour as an initial condition to further improve convergence and identifiability. It can accurately recover latent variables in spatial and non-spatial tasks. When applied to a large hippocampal dataset SIMPL converges on smaller, more numerous, and more uniformly sized place fields than those based on behaviour, suggesting the brain may encode space with greater resolution than previously thought.

## 1 Introduction

Large neural populations in the brain are known to encode low-dimensional, time-evolving latent variables which are, oftentimes, closely related to behaviour (Afshar et al., 2011; Harvey et al., 2012; Mante et al., 2013; Carnevale et al., 2015). Coupled with the advent of modern neural recording techniques (Jun et al., 2017; Wilt et al., 2009) focus has shifted from single-cell studies to the joint analysis of hundreds of neurons across long time windows, where the goal is to extract latents using a variety of statistical (Yu et al., 2008a; Cunningham & Yu, 2014; Kobak et al., 2016; Zhao & Park, 2017; Williams et al., 2020; Bjerke et al., 2023) and computational (Van der Maaten & Hinton, 2008; Pandarinath et al., 2018; Mackevicius et al., 2019) methods.

This paradigm shift is particularly pertinent in mammalian spatial memory and motor systems where celebrated discoveries have identified cells whose neural activity depends on behavioural variables such as position (O'Keefe & Dostrovsky, 1971; Hafting et al., 2005), heading direction (Taube et al., 1990), speed (McNaughton et al., 1983), distance to environmental boundaries/objects (Lever et al., 2009; Høydal et al., 2019) and limb movement direction(Georgopoulos et al., 1986) through complex and non-linear tuning curves. Characterising neural activity in terms of behaviour remains a cornerstone practice in these fields however the implicit assumption supporting it — that the latent variable encoded by neural activity *is and only is* the behavioural variable — is increasingly being called into question (Sanders et al., 2015; Whittington et al., 2020; George et al., 2024b).

The brain is not a passive observer of the world. The same neurons which encode an animal's *current* position/behavioural state are also used to plan a future routes (Spiers & Maguire, 2006), predict upcoming states (Muller & Kubie, 1989; Mehta et al., 1997; Stachenfeld et al., 2017) or recall/"replay" past positions (Squire et al., 2010; Carr et al., 2011), necesarily causing the encoded latent variables to deviate from behaviour. Nor is the brain a perfect observer; uncertainty due to limited, noisy or ambiguous sensory data can lead to similar discrepancies. Measurement inaccuracies can contribute further. These hypotheses are supported by analyses which show that it is rarely, if ever, possible to perfectly decode "behaviour" from neural data (Glaser et al., 2020) and the observation that neurons show high variability under identical behavioural conditions (Fenton & Muller, 1998; Low et al., 2018). All combined, these facts hint at a richer and more complex internal neural code. When this is not accounted for tuning curves will be blurred, distorted or mischaracterised relative to their true form. For example, consider an animal situated at position X 'imagining' or 'anticipating' a remote position, Y, for which a place cell is tuned. This might trigger the cell to fire leading to the mistaken conclusion that the cell has a place field at location X.

Nonetheless, the fact that behaviour is often a close-but-imperfect proxy for the true latent motivates searching for techniques which *exploit* this link. Most existing methods for latent discovery don't exploit behaviour (Gao et al., 2016; Gondur et al., 2023) at the cost of complexity and interpretability. Others don't model temporal dynamics(Zhou & Wei, 2020; Schneider et al., 2023; Lawrence, 2003), don't scale to large datasets (Wang et al., 2005; Nam, 2015; Wu et al., 2017), can't model complex non-linear tuning curves (Pandarinath et al., 2018; Hurwitz et al., 2021; Duncker et al., 2019; Linderman et al., 2016; Gondur et al., 2023), or aren't designed for spiking datasets(Lawrence, 2003; Krishnan et al., 2015). Moreover, many of these methods are conceptually complex, lack usable code implementations, or necessitate GPUs limiting their accessibility.

**Contributions** Here we introduce SIMPL (Scalable Iterative Maximisation of Population-coded Latents), a straightforward yet effective enhancement to the current paradigm. Our approach fits tuning curves to observed behaviour and refines these by iterating a two-step process. First the latent trajectory is *decoded* from the current tuning curves then, the tuning curves are *refitted* based on this decoded latent trajectory. SIMPL imposes minimal constraints on the tuning curve structure, scales well to large datasets without relying on neural networks which can be expensive to train. Theoretical analysis establishes formal connections to expectation-maximisation (EM, Dempster et al. 1977) for a flexible class of generative models. By exploiting behaviour as an initialisation, SIMPL converges fast and helps mitigate local minima and identifiability (Hyvärinen & Pajunen, 1999; Locatello et al., 2019) issues. This allows it to reliably return refined tuning curves and latents which remain close to, but improve upon, their behavioural analogues readily admitting direct comparison. All in all, SIMPL is able to identify temporally smooth latents and complex tuning curves related to behaviour, while remaining cheap and natively supporting spiking data — a distinguishing set of features in the field of latent variable models for neural data analysis.

We first validate SIMPL on a dataset of synthetically generated 2D grid cells. Next, we apply SIMPL to rodent electrophysiological hippocampal data (Tanni et al., 2022) and show it modifies the latent space in an incremental but significant way: optimised tuning curves are better at explaining held-out neural data and contain sharper, more numerous place fields allowing for a reinterpretation of previous experimental results. Finally, we apply SIMPL to somatosensory dataset for a monkey performing a centre-out reaching task (Chowdhury et al., 2020). SIMPL, with a 4D latent space, provides a good account of the data with the latent variables initialised to (and remaining correlated with) the monkeys hand-position and hand-velocity. With only two hyperparameters, SIMPL can be run quickly on large neural datasets [1] without requiring a GPU. It outperforms popular alternative techniques based on neural networks (Schneider et al., 2023; Zhao & Park, 2017) or Gaussian processes(Lawrence, 2003; Wang et al., 2005) and is over $15\times$ faster. This makes it a practical alternative to existing tools particularly of interest to navigational or motor-control communities where abundant data is explained well by measurable behaviours (position, hand dynamics). We provide an open-source JAX-optimised (Bradbury et al., 2018) implementation of our code[2].

---

[1] One-hour recordings of 200 neurons ($10^6$ spikes) takes 1 minute to run on a CPU laptop.

[2] Code and a demo can be found at: `https://github.com/TomGeorge1234/SIMPL`

## 2 METHOD

Here we give a high-level description of the SIMPL algorithm. Comprehensive details and a theoretical analysis linking SIMPL to expectation-maximisation, are provided in the Appendix.

### 2.1 THE MODEL

SIMPL models *spike trains* of the form $\mathbf{s} := (s_{ti})_{t=1,\dots T}^{i=1,\dots N}$, where $s_{ti}$ represents the number of spikes emitted by neuron $i$ between time $(t-1) \cdot dt$ and $t \cdot dt$. We denote $\mathbf{s}_t := (s_{t1}, \dots, s_{tN})$ the vector of spike counts emitted by all neurons in the t-th time bin. SIMPL posits that such spike trains $\mathbf{s}$ are modulated by a *latent, continuously-valued, low-dimensional, time-evolving* variable $\mathbf{x} := (\mathbf{x}_t)_{t=1,\dots,T} \in \mathbb{R}^D$ through the following random process:

$$\mathbf{x}_{t+1} \mid \mathbf{x}_t \sim \quad \mathcal{N}(\mathbf{x}_t, \sigma_v^2 \mathbf{I}) \qquad \text{(Latent dynamics)} \qquad (1)$$

$$s_{ti} \mid \mathbf{x}_t \sim \quad \text{Poisson}(f_i(\mathbf{x}_t)) \qquad \text{(Emission model)} \qquad (2)$$

where $\sigma_v := \mathrm{v} \cdot \mathrm{d}t$ and $\mathbf{x}_0 \sim \mathcal{N}(0, \sigma_0^2 \mathbf{I})$. This generative model enforces a tunable (through the velocity hyperparameter $v$) amount of temporal smoothness in the trajectories. At each time step the latent variable $\mathbf{x}_t$ determines the instantaneous firing rate of all neurons via their intensity functions $f_i$ (hereon called *tuning curves*, collectively denoted $\mathbf{f}$), which are unknown a priori, and which SIMPL will estimate. Moreover, we make the common assumption that all neurons are *conditionally independent* given $\mathbf{x}_t$, i.e. $p(\mathbf{s}_t|\mathbf{x}_t) = \prod_{i=1}^N p(s_{ti}|\mathbf{x}_t)$. Finally, we assume the latent variable $\mathbf{x}$ is Markovian, a common assumption in the neuroscience literature. This model has been previously studied in the literature (Smith & Brown, 2003; Macke et al., 2011), albeit using highly restrictive tuning curve models, something which SIMPL avoids.

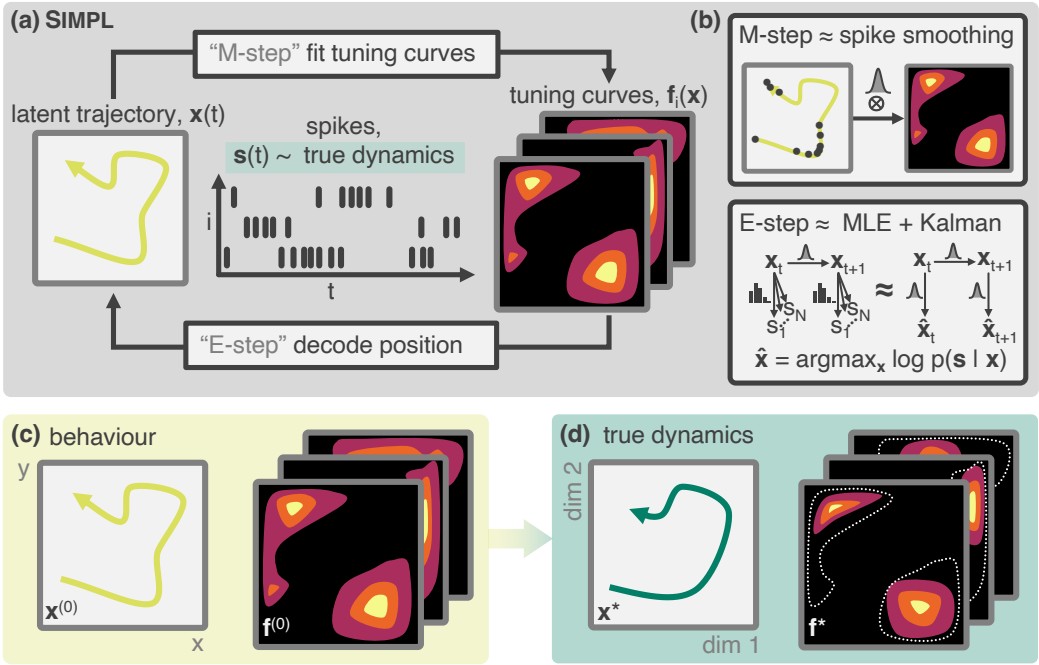

Figure 1: Schematic of SIMPL. **(a)** A latent variable model (LVM) for spiking data $(\mathbf{f}_i(\mathbf{x}), \mathbf{x}(t))$ is optimised by iterating a two-step procedure closely related to the expectation-maximisation: First, tuning curves are fitted to an initial estimate of the latent trajectory (an "M-step"). The latent is then *re*decoded from these tuning cuves (an "E-step"). **(b)** SIMPL fits tuning curves using a kernel-smoothed estimate (top) and decodes the latent variables by Kalman-smoothing maximum likelihood estimates. **(c)** Measured behaviour is used to initialise the algorithm as it is often closely related to the true generative LVM **(d)**.

### 2.2 THE SIMPL ALGORITHM

**Outline** We now seek an estimate of the true, unknown latent trajectory $\mathbf{x}^\star$ and tuning curves $\mathbf{f}^\star$ that led to an observed spike train, $\mathbf{s}$. SIMPL does so by iterating a two-step procedure closely related to

the expectation-maximisation (EM) algorithm: first, tuning curves are fitted to an initial estimate of the latent variable (the "M-step"), which are then used to decode the latent variable (the "E-step"). This procedure is then repeated using the new latent trajectory, and so on until convergence.

**The M-step** In the M-step (or "fitting" step) of the $e$-th iteration SIMPL fits tnng curves to the current latent trajectory estimate $\mathbf{x}^{(e)}$ using a smooth, kernel-based estimate

$$f_i^{(e)}(\mathbf{x}) := \frac{\sum_{t=1}^T s_{ti}\, k(\mathbf{x}, \mathbf{x}_t^{(e)})}{\sum_{t=1}^T k(\mathbf{x}, \mathbf{x}_t^{(e)})} \approx \frac{\text{\# spikes at x}}{\text{\# visits to x}} \tag{3}$$

for some kernel $k$. In practice, we use a Gaussian kernel with small bandwidth $\sigma$. Such a tuning curve model is conceptually simple and free from the optimisation, misspecification or interpretability issues of most parametric models. It constitutes a notable departure from alternatives which use a neural network (Zhou & Wei, 2020; Schneider et al., 2023) to model tuning curves and is particularly well suited to low-dimensional latent spaces.

**The E-step** In the E-step SIMPL seeks to infer (or "decode") a new estimate of the latent from the spikes and current tuning curves, $\mathbf{x}^{(e+1)} = \mathbb{E}_{p(\mathbf{x}|\mathbf{s},\mathbf{f}^{(e)})}[\mathbf{x}]$. Directly performing this inference from the spikes is difficult due to the non-linearity and non-Gaussianity of the emission model in Eq (2). Instead, SIMPL first calculates the *maximum likelihood estimate* (MLE) of $\mathbf{x}$, denoted $\widehat{\mathbf{x}}$. Then, by making a linear-Gaussian approximation to $p(\widehat{\mathbf{x}}_t|\mathbf{x}_t) \approx \mathcal{N}(\mathbf{x}_t; \boldsymbol{\Sigma}_t)$, the variables $(\mathbf{x}, \widehat{\mathbf{x}})$ form a Linear Gaussian State Space Model (LGSSM) fully characterised by $\sigma_v^2 \mathbf{I}$ (the transition noise covariance) and $\boldsymbol{\Sigma}_t$ (the observation noise covariance). This enables efficient inference via *Kalman smoothing* of the MLEs in order to approximate $\mathbf{x}^{(e+1)} = \mathbb{E}_{p(\mathbf{x}|\widehat{\mathbf{x}})}[\mathbf{x}]$ (schematic in Fig. 1b).

$$\widehat{\mathbf{x}}^{(e+1)} := \arg\max_{\mathbf{x}} \log p(\mathbf{s}|\mathbf{x}, \mathbf{f}^{(e)})$$
$$\mathbf{x}^{(e+1)} := \mathbb{E}_{p(\mathbf{x}|\widehat{\mathbf{x}}^{(e+1)})}[\mathbf{x}] \approx \text{KalmanSmooth}(\widehat{\mathbf{x}}^{(e+1)}; \sigma_v^2 \mathbf{I}, \boldsymbol{\Sigma}_t) \tag{4}$$

Crucially, the linear-Gaussian approximation is *not* made on the spiking emissions $p(\mathbf{s}|\mathbf{x})$, which is non-linear and non-Gaussian by design, but on $p(\widehat{\mathbf{x}}|\mathbf{x})$, a quantity which is provably asymptotically Gaussian in the many-neurons regime (theoretical argument and an explicit formula for $\boldsymbol{\Sigma}_t$ in B.1).

**Behavioural initialisation** Spike trains often come alongside behavioural recordings $\mathbf{x}^b$ thought to relate closely to the true latent variable $\mathbf{x}^b \approx \mathbf{x}^\star$. SIMPL leverages this by setting the initial decoded latent trajectory, to measured behaviour $\mathbf{x}^{(0)} \leftarrow \mathbf{x}^b$. We posit that *behavioural initialisation* will place the first iterate of SIMPL within the vicinity of the true trajectory and tuning curves, accelerating convergence and favouring the true latent and tuning curves $(\mathbf{x}^\star, \mathbf{f}^\star)$ over alternative isomorphic pairs $(\phi(\mathbf{x}^\star), \mathbf{f}^\star \circ \phi^{-1})$ whose latent space is *warped* by an invertible map $\phi$ but which would explain the data equally well. This amounts to an inductive bias favouring tuning curves close to those calculated from behaviour. Through ablation studies we confirm these beneficial effects.

All in all, SIMPL is interpretable and closely matches common practice in neuroscience (e.g. kernel-based curve fitting, MLE-based decoding); moreover, it can be formally related to a generalised version of the EM-algorithm, for which theoretical guarantees may be obtained. We leave to the appendix detailed theoretical arguments justifying the validity of SIMPL as well as its connection to EM.

---

**Algorithm 1** SIMPL: An algorithm for optimizing tuning curves and latents from behaviour

---

1: $\mathbf{s} \in \mathbb{N}^{N \times T}$           $\triangleright$ Spike count matrix
2: $\mathbf{x}^{(0)} \in \mathbb{R}^{D \times T}$        $\triangleright$ Initial latent estimate e.g. measured position of animal
3: **procedure** SIMPL($\mathbf{s}, \mathbf{x}^{(0)}$)
4:     **for** $e \leftarrow 0$ to $E$ **do**          $\triangleright$ Loop for $E$ iterations
5:         $\mathbf{f}^{(e)} \leftarrow$ FitTuningCurves($\mathbf{x}^{(e)}, \mathbf{s}$)        $\triangleright$ The "M-step"
6:         $\mathbf{x}^{(e+1)} \leftarrow$ DecodeLatent($\mathbf{f}^{(e)}, \mathbf{s}$)        $\triangleright$ The "E-step"
7:     **end for**
8:     **return** $\mathbf{x}^{(E+1)}, \mathbf{f}^{(E)}$        $\triangleright$ The optimised latent and tuning curves
9: **end procedure**

---

## 3 RESULTS

### 3.1 CONTINUOUS SYNTHETIC DATA: 2D GRID CELLS

First we tested SIMPL on a realistic navigational task by generating a large artificial dataset of spikes from a population of $N = 225$ 2D grid cells — a type of neuron commonly found in the medial entorhinal cortex (Hafting et al., 2005) — in a 1 m square environment. All grid cells had a maximum firing rate of 10 Hz and were arranged into three discrete modules, 75 cells per module, of increasing grid scale from 0.3–0.8 m (Fig. 2c). A latent trajectory, $\mathbf{x}^\star$, was then generated by simulating an agent moving around the environment for 1 hour under a smooth continuous random motion model. Data was sampled at a rate of 10 Hz giving a total of $T = 36,000$ time bins ($\sim$ 800,000 spikes). All data was generated using the `RatInABox` package (George et al., 2024a).

The initial trajectory, $\mathbf{x}^{(0)}$, was generated by adding smooth Gaussian noise to the latent such that, on average, the true latent and initial condition differed by 20 cm (Fig. 2a, top panel). This discrepancy models the agent's internal position uncertainty and/or a measurement error. It sufficed to obscure almost all structure from the initial tuning curves $\mathbf{f}^{(0)}(\mathbf{x})$ (Fig. 2b, top). To assess performance we track to the log-likelihood of training and test spikes (see Appendix C.5 for how we partition the dataset). We also calculate the error between the true and latent trajectory the epoch-to-epoch change in the tuning curves and the negative entropy (hereon called "spatial info") of the normalized tuning curves as a measure of how spatially informative they are (Fig. 2d).

SIMPL was then run for 10 epochs (total compute time 39.8 CPU-secs on a consumer grade laptop). The true latent trajectory and receptive fields were recovered almost perfectly and the log-likelihood of both train and test spikes rapidly approached the ceiling performance with negligible overfitting. As expected, SIMPL performs better on larger datasets, Fig. 2e, however performance remains good even with substantially smaller datasets (e.g. 50 cells for a duration of 5 minutes). We also swept across the velocity and kernel bandwidth hyperparameters ($v, \sigma$) and found SIMPL was surprisingly robust to changes in these hyperparameters within reasonable limits (see D.2).

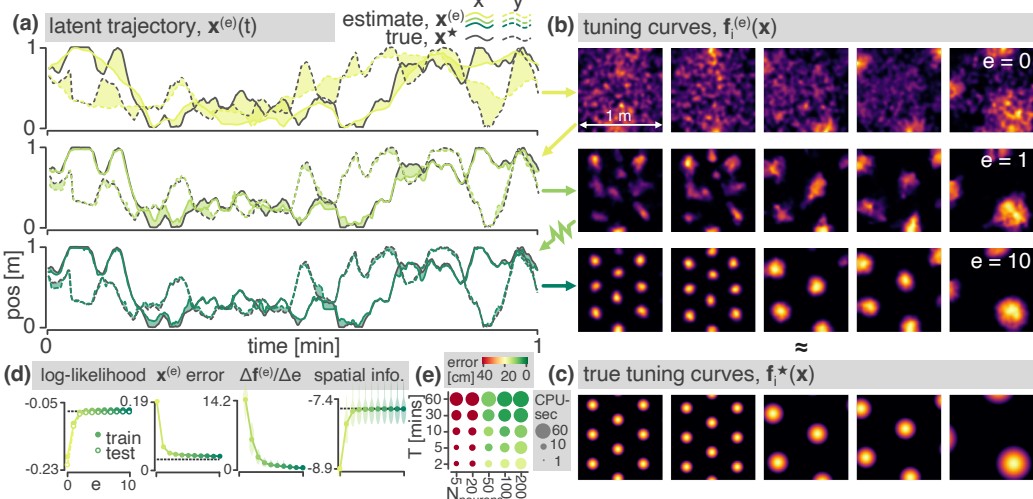

Figure 2: Results on a synthetic 2D grid cell dataset. **(a)** Estimated latent trajectories (epochs 0, 1 and 10). Initial conditions are generated from the true latent (black) by the addition of slow Gaussian noise. Shaded zones show the discrepancy between the true and estimated latent. **(b)** Tuning curve estimates for 5 exemplar grid cells. **(c)** Ground truth tuning curves. **(d)** Performance metrics: *Left:* log-likelihood of the training and test spikes (averaged per time step, dotted line shows ceiling performance on a model initialised with the true latent). *Middle-left:* Euclidean distance between the true and estimated latent trajectories (averaged per time step). *Middle-right:* Epoch-to-epoch change in the tuning curves showing they stabilise over iteration. *Right:* Cell spatial information. Violin plots, where shown, give distributions across all neurons. **(e)** A sweep over the number of cells and the duration of the trajectory.

Finally, despite having an implicit prior for temporally-smooth latent dynamics, further synthetic analysis revealed SIMPL is still able recover *discontinuous* latent trajectories (for example those

containing jumpy-like "replay" events, see Appendix D.3) or even *discrete* latents in a non-dynamical task akin to a discrete two-alternative forced choice task (2AFC, see Appendix D.1).

## 3.2 HIPPOCAMPAL PLACE CELL DATA

Having confirmed the efficacy of SIMPL on synthetic data, we next tested it on real dataset of hippocampal neurons recorded from a rat as it foraged in a large environment (Tanni et al., 2022). This dataset consists of $N = 226$ neurons recorded over 2 hours, binned at 5 Hz giving $T = 36,000$ data samples and $\sim 700,000$ spikes. Many of these cells are place cells (O'Keefe & Dostrovsky, 1971) which, in large environments, are known to have multiple place fields (Park et al., 2011).

We initialised with the animal's position, as measured by an LED located between its ears, and optimised for 10 epochs. The log-likelihood of test and train spikes both increased, converging after 4 epochs (Fig. 3b) in a compute time of $\sim 40$ CPU-secs. We then analysed the shapes and statistics of the tuning curves: After optimisation, tuning curves were visibly sharper, Fig. 3a; previously diffuse place fields contracted (e.g. the third exemplar tuning curve) or split into multiple, smaller fields (second exemplar). Occasionally, new place fields appeared (fourth exemplar) or multiple place fields merged into a single larger field (fifth exemplar). Statistically, tuning curves had significantly more individual place fields (+19%, mean 1.14→1.41 per cell, $p = 0.0035$ Mann Whitney U tests), substantially higher maximum firing rates (+45%, median 4.2→6.1 Hz, $p = 9.8 \times 10^{-7}$) and were more spatially informative ($p = 0.038$). Individual place fields became smaller (-25%, median 0.59→0.44 m$^2$) and rounder (+8%, median 0.63→0.68, $p = 0.0037$).

To ensure these observed changes weren't merely an artefact of the optimisation procedure we generated a control dataset by resampling spikes from the behaviour-fitted tuning curves, $\mathbf{s}_{con} \sim p(\cdot|\mathbf{x}^{(0)}, \mathbf{f}^{(0)})$. Control spikes thus had very similar temporal statistics and identical tuning curves to those in the hippocampal dataset but, critically, were generated from a known ground truth model exactly equal to their initialization. Thus, any changes in the control tuning curves post-SIMPL must be artefactual. Indeed, no significant changes were observed besides a slight *increase* in field area (Fig. 3bc, grey) providing strong evidence the significant changes observed in the real data (e.g. the *decrease* in field area) were genuine, reflecting the true nature of hippocampal tuning curves.

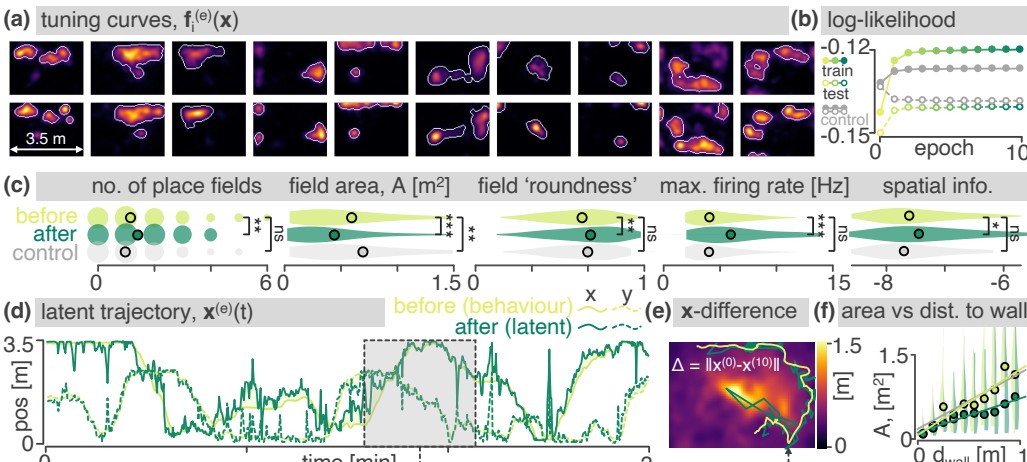

Figure 3: Results on a hippocampal place cell dataset collected by Tanni et al. (2022). **(a)** Exemplar tuning curves before and after optimization. Automatically identified place field boundaries shown in white. **(b)** Log-likelihood of test and train spikes. Control model shown in grey. **(c)** Statistics analysis of place fields. Violin plots show the distributions over all place fields / cells. **(d)** The final latent trajectory estimated from SIMPL (green) overlaid on top of the measured position of the animal (used as initial conditions, yellow). **(e)** Behavioural discrepancy map: the average discrepancy between the latent and behaviour as a function of the optimised latent $\mathbf{x}^{(10)}$. Overlaid is a snippet of the behavioural vs optimised true latent trajectory. **(f)** Place field area as a function of the distance to the nearest wall.

The optimized latent trajectory $\mathbf{x}^{(10)}$ remained highly correlated with behaviour ($R^2 = 0.86$, Fig. 3d) occasionally diverging for short periods as it "jumped" to and from a new location, as if the

animal was mentally teleporting itself (an example is visualized in Fig. 3e). We calculated the difference between the optimised latent and the behaviour at each time point, $\Delta_t = \|\mathbf{x}_t^{(0)} - \mathbf{x}_t^{(10)}\|_2$, and visualized this as a heat map overlaid onto the latent space (Fig. 3e). We found that the latent discrepancy was minimal near the edges of the environment and peaked near the centre, perhaps because sensory input is scarce in the centre of the environment due to fewer visual and tactile cues.

Tanni et al. (2022) observed that the size of a place field size increases with its distance to a wall. Our observation—that the latent discrepancy is highest in the centre of the environment—suggests one possible hypothesis: behavioural place fields merely *appear* larger in the centre of the environment because they are blurred by the correspondingly larger latent discrepancy. If true, this trend should weaken after optimisation, once the "true" latent has been found.

To test this we plotted field size against distance-to-wall (Fig. 3f); optimized fields, like behavioural fields, were small very near to the walls and grew with distance (replicating the result of Tanni et al. (2022)), but this correspondence stopped after $\sim 0.5$ m beyond which the optimized place fields size grew more weakly with distance-to-wall. This supports our hypothesis, suggesting a substantial fraction of the correlation between size and distance isn't a fundamental feature of the neural tuning curves but an artefactual distortion in the tuning curves, something which can be corrected for using SIMPL.

### 3.3 SOMATOSENSORY CORTEX DATA DURING A HAND-REACHING TASK

To test SIMPL beyond navigational/hippocampal datasets we ran it on a macaque somatosensory cortex dataset Chowdhury et al. (2020). During this recording a monkey made a series of reaches to a target in one of 8 directions, 4. On half of the trials the reach was "active" whereby the monkey moved the manipulandum towards the target by itself. On the other half, the reach was "passive", whereby the monkey's hand was bumped in the direction of one of the targets by a force applied to the manipulandum, forcing the monkey to correct and return the cursor to the centre. We binned the data ($N = 65$ neurons, 37 mins, $\sim 10^6$ spikes) at 20 Hz and ran SIMPL models on its entirety (i.e. active and passive reaches, as well as the inter-trial intervals) for 10 epochs.

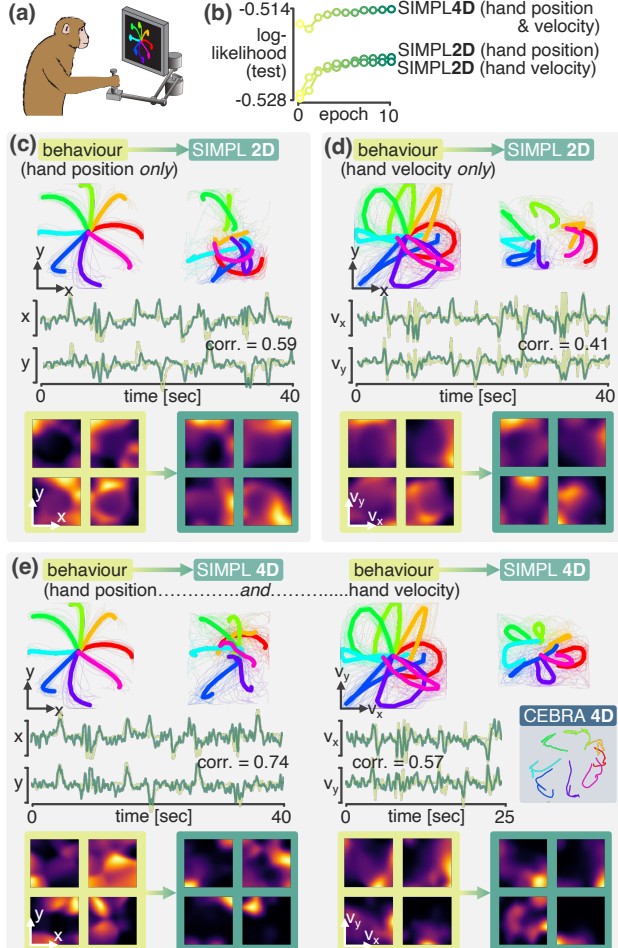

Figure 4: SIMPL applied to somatosensory cortex data. **(a)** A macaque performs centre-out reaches; $N = 65$ somatosensory neurons are recorded. **(b)** Log-likelihood curves for the three SIMPL models in panels c–e. **(c)** SIMPL trained with a 2D latent initialised from hand position. Top-left: raw behaviour, averaged across trials aligned to movement onset; top-right: after SIMPL. Middle: 40 s of behaviour (yellow) and latent (green). Bottom: exemplar tuning curves before and after SIMPL. **(d)** As in c, but initialised with hand velocity. **(e)** As in c, but with a 4D latent initialised to hand-position (dims 1 and 2) and velocity (dims 3 and 4). Inset: 2D visualisation of a 4D latent embedding from CEBRA trained on hand position, adapted from Schneider et al. (2023).

First SIMPL was run with a 2D latent initialised to the monkeys measured x- and y-hand position (Fig. 4c). Afterwards, the latent trajectory—here averaged across trials with the same direction, aligned to movement onset—had diverged from, but remained correlated with, initial hand-position

(correlation = 0.59). Despite an improvement in likelihood over the behavioural initialisation, latent trajectories for distinct directions substantially overlapped with one another, indicating an insufficient dimensionality to capture the full complexity of the data. A similar result was obtained when initialising to hand-velocity (Fig. 4d).

We then trained SIMPL with a 4D latent space. Two of the dimensions were initialised with hand position and the other two with hand velocity. This model performed better than either 2D model, converging to a higher likelihood. The latent dimensions initialised to hand-position remained highly correlated with hand-position (corr. = 0.74) after optimisation as did the velocity dimensions (corr. = 0.57). The latent trajectory was also more structured, with distinct and less overlapping motifs for each trial type. We visualised two-dimensional slices of the four-dimensional tuning curves for each neuron and found that they had well-defined receptive fields, similar to place fields in the hippocampus, which were visibly sharper after optimisation. These results suggest that the somatosensory cortex neurons encode a complex and high-dimensional latent, closely correlated to hand position and velocity, which can be partially recovered by SIMPL.

### 3.4 PERFORMANCE IS IMPROVED BY INITIALISING AT BEHAVIOUR

Latent variable models trained with EM can experience two issues that usually complicate the scientific interpretability of their results. The first concerns the *quality* of the solution; does the algorithm converge on a good model of the data which predicts the spikes well? The second issue concerns *identifiability*; even if the recovered latent trajectory and tuning curves $(\mathbf{f}^{(e)}, \mathbf{x}^{(e)})$ are of high quality, they may differ from the true ones $(\mathbf{f}^{\star}, \mathbf{x}^{\star})$ by some invertible "warp" $\phi$ in a way that does not affect the overall goodness-of-fit of the model. These warps could include innocuous rotations and symmteries or, more concerningly if the exact structure of the tuning curve is a quantity of interest, stretches or fragmentations. Here we show that behavioural initialisation drastically minimises the severity of both of these issues for SIMPL.

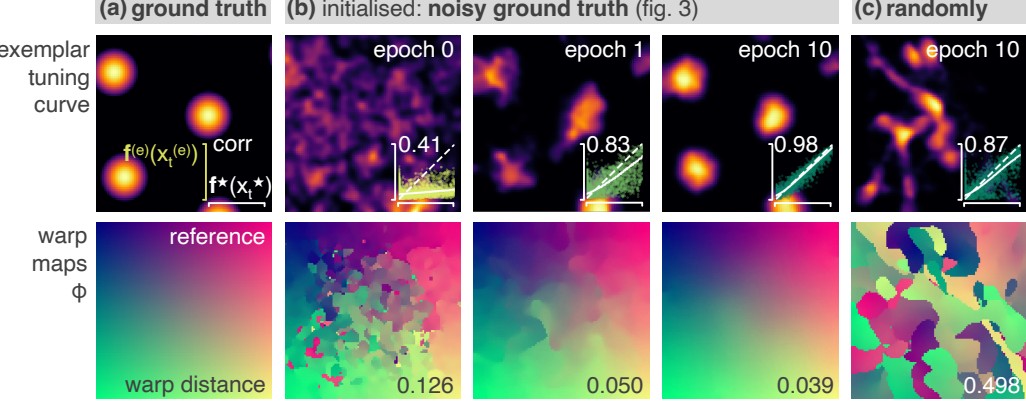

Figure 5: Latent manifold analysis: **(Top)** Examplar tuning curve in **(a)** the ground truth latent space, **(b)** the latent space discovered by behaviourally-initialised-SIMPL after 0, 1 and 10 epochs and **(c)** the latent space discovered by SIMPL initialised with a random latent trajectory. Inset scatter plots show the true and predicted firing rates of all neurons across all times as well as their correlation values ("accurate" models have higher correlations). **(Bottom)** The warp mappings from each latent space to the "closest" location in ground truth as measured by the distance between the tuning curves population vectors.

To do so, we first assess the absolute goodness-of-fit of SIMPL by computing, for all neurons, the correlation between the estimated instantaneous firing rates $f_i^{(e)}(\mathbf{x}_t^{(e)})$ (a quantity invariant to warping) and the true firing rates $f_i^{\star}(\mathbf{x}_t^{\star})$. SIMPL converges to a highly accurate model (r=0.98) under behavioural initialization, but to a less accurate, though still quite accurate, model ($r = 0.87$) when initialised with a random trajectory uncorrelated to the true latent. Next, we estimate, quantify and visualize the warp map $\phi$ between SIMPL's estimates $(\mathbf{f}^{(e)}, \mathbf{x}^{(e)})$ and the ground truth $(\mathbf{f}^{\star}, \mathbf{x}^{\star})$. We obtain this by finding, for every location in the warped space, the position in the true latent space where the tuning curves are most similar ($\phi(\mathbf{x}) = \arg\min_{\mathbf{y}} \|\mathbf{f}^{\star}(\mathbf{y}) - \mathbf{f}^{(e)}(\mathbf{x})\|_2$). We then quantify the "warpness" of this mapping as the average distance between $\mathbf{x}$ and $\phi(\mathbf{x})$ across the environment, normalized by its characteristic length scale (1 m). This warp-distance should be 0 for totally un-

warped models and $\mathcal{O}(1)$ for heavy warps. In addition to perfectly fitting the data, the solution found by SIMPL under behavioural initialization is minimally warped (warp dist = 0.050). In contrast, the good (but imperfect) solution found by SIMPL under random initialization is very heavily warped (warp dist. = 0.498) in a fragmented manner. These results (Fig. 5) strongly motivate the use of behavioural initializations in latent variable models as an effective means to encourage convergence towards latent spaces which are both accurate and un-warped with respect to the ground truth.

### 3.5 BENCHMARKING SIMPL AGAINST EXISTING TECHNIQUES

We compared SIMPL to four popular methods for latent variable extraction: pi-VAE (Zhou & Wei, 2020), CEBRA (Schneider et al., 2023) (which use neural network function approximators), GPLVM (Lawrence, 2003) and GPDM (Wang et al., 2005) (which use Gaussian processes). Crucially, and like SIMPL, none of these methods make restrictive linear assumptions about the structure of the tuning curves.

To match SIMPL, we initialise the latent variable estimates of GPLVM and GPDM to behaviour (pi-VAE and CEBRA handle behaviour natively by using it to condition a prior over the latent or as a contrastive label). All models were trained for their default number of iterations/epochs. After training we aligned the discovered latents to behaviour and visualised them on top of the ground truth (Fig. 6c). All models successfully uncovered a latent trajectory closer to the ground truth than behaviour (Fig. 6b). SIMPL performed better than the other models, achieving a final error of 4.2 cm, half that of pi-VAE (8.4 cm).

We posit that pi-VAE, CEBRA and GPLVM may suffer from the lack of an explicit dynamical systems component in their generative models while GPDM may suffer from the data-subsampling we were required to do to cap the training time to less than two-hours. SIMPL converged in 40 seconds, over 15 times quicker than the next fastest (pi-VAE, 10.4 minutes, Fig. 6a). Except for GPDM, which required a GPU, all techniques were run and timed on a CPU. Only SIMPL was able to recover sharp and accurate grid fields close to the ground truth.

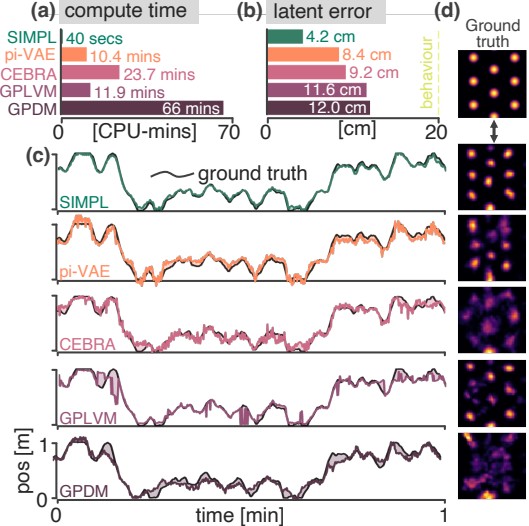

## 4 RELATED WORK

Probabilistic inference in neural data modulated by latent variables has been a major topic of study for decades — see, e.g. Tipping & Bishop (1999); Yu et al. (2006; 2008b;a); Macke et al. (2011); Mangion et al. (2011); Park et al. (2015); Gao et al. (2016); Hernandez et al. (2018); Dong et al. (2020); Zhou & Wei (2020); Gondur et al. (2023); Bjerke et al.

Figure 6: Comparison to pi-VAE, CEBRA, GPLVM and GPDM on the synthetic grid cell dataset. **(a)** Compute time. **(b)** Final error in the latent. **(c)** Alignment of the discovered latent to the ground truth. **(d)** Exemplar tuning curves constructed using kernel-based estimation on the latent (i.e. an "M-step").

(2023) — however not all methods were designed for the kind of data considered in this work. Many methods contain model complex latent space dynamics but combine these with simplistic tuning curves which restrict firing rates to (exponential-)linear functions of the latent (Smith & Brown, 2003; Yu et al., 2008a; Macke et al., 2011; Duncker et al., 2019; Linderman et al., 2016; Pandarinath et al., 2018; Zoltowski et al., 2020; Sani et al., 2021; Hurwitz et al., 2021; Kim et al., 2021; Gondur et al., 2023) so cannot interpretably account for the representations (place cells, grid cells) considered here. Other methods do not/cannot use behaviour to aid latent discovery (Gao et al., 2016; Nam, 2015; Hernandez et al., 2018; Gondur et al., 2023; Bjerke et al., 2023) instead taking a fully "unsupervised" approach (meaning they can be applied to spike data without an obvious behavioural correlate) at the expense of complexity and identifiability.

Algorithms that both don't restrict to simplistic linear tuning curves and exploit behaviour form a small set of relevant alternatives to SIMPL. Behaviour-informed latent discovery tools have become

popular in recent years due to the explosion of large neural datasets taken from behaving animals and the observation that behaviour can explain substantial variance in the neural dynamics.

Gaussian process latent variable models (GPLVMs), Lawrence (2003); Wang et al. (2005) form a family of methods that learn smooth, non-linear tuning curves by placing GP priors on them and performing approximate marginal log-likelihood optimisation on the latent variable. Popular implementations leave the initial condition of this optimisation user-defined and therefore compatible with the behaviour-informed initialisation used here. However, most such models were introduced outside of the neuroscience literature thus use Gaussian (instead of Poisson) emission models (Lawrence, 2003; Wang et al., 2005; Jensen et al., 2020), or do not make smoothness assumptions on the latent trajectory (Jensen et al., 2020; Lawrence, 2003). P-GPLVM, which employs Poisson emissions and a GP prior on the latent trajectory, is an exception, but its cubic scaling with time points makes it impractical for hour-long datasets. In contrast, available GPLVM implementations(Bingham et al., 2018) use inducing point approximations to achieve linear time complexity.

CEBRA (Schneider et al., 2023) learns a deterministic neural network mapping from spikes to latents using behaviour- or time-guided contrastive learning. Unlike most methods, CEBRA does not natively learn a generative model nor tuning curves, which are of primary interest in our setting. CEBRA also treats each data point independently instead of modelling whole-trajectories preventing it from taking advantage of the temporal smoothness inherent in many underlying latent codes.

pi-VAE (Zhou & Wei, 2020) uses a variational autoencoder (Kingma & Welling, 2014) to infer the latent trajectories and learn tuning curves using neural network function approximators. pi-VAE places a learnable prior, conditioned on behaviour, to the latent variable in order to obtain a model with provable identifiability properties. However, pi-VAE suffers from the same limitation as CEBRA in that it treats each data point as an i.i.d observation instead of a part of a whole trajectory.

The properties of large scale neural datasets suggest five desiderata on the algorithms used to analyse them. These are (1) the absence of restrictive tuning curve assumptions, (2) modelling smooth latent dynamics, (3) the presence of a spiking component (e.g. Poisson emissions), (4) the ability to exploit behaviour (including as an initial condition) and (5) scalability to large datasets. None of the methods described in our literature review satisfy all five desiderata. In Appendix E we provide a table comparing all methods discussed in this section and more with respect to these desiderata.

## 5 DISCUSSION

We introduced SIMPL, a tool for optimizing tuning curves and latent trajectories using a technique which refines estimates obtained from behaviour. It hinges on two well-established sub-routines — tuning curve fitting and decoding — which are widely used by both experimentalists and theorists for analysing neural data. By presenting SIMPL as an iterative application of these techniques, we aim to make latent variable modelling more accessible to the neuroscience community.

SIMPL could be seen as an instance of a broader class of latent optimization algorithms. In principle *any* curve fitting procedure and *any* decoder (which uses those tuning curves) could be coupled into a candidate algorithm for optimizing latents from neural data. Our specific design choices, while attractive due to their conceptual and computational simplicity, will come with limitations. For example, we predict SIMPL's kernel-based estimator won't scale well to very high dimensional latent spaces (Györfi et al., 2006) where parametric models, e.g. a neural networks, are known to perform better (Bach, 2017), potentially at the cost of compute time.

Our synthetic analysis focused on settings where behaviour and the true latent differed only in an unbiased manner. It would be interesting to determine if SIMPL's performance extends to more complex perturbations. Fast, non-local and asymmetric perturbations are common in the brain; for instance "replay"(Carr et al., 2011) where the latent jumps to another location in the environment. Likewise, during theta sequences(Maurer et al., 2006), the encoded latent moves away from the agent. This forward-biased discrepancy could theoretically induce a backward-biased skew in behavioural place fields, even if the true tuning curves remain unskewed. If this is the case, proper latent dynamical analysis—via tools like SIMPL—could help reinterpret the predictive nature of place field tuning curves (Stachenfeld et al., 2017; Fang et al., 2023; Bono et al., 2023; George et al., 2023), similar to how it reduced the asymmetry in place field sizes further from walls (Fig. 3f).

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

# Supplementary Material for "SIMPL: Scalable and hassle-free optimisation of neural representations from behaviour"

## A  BACKGROUND

### A.1  EXPECTATION MAXIMIZATION

Expectation Maximization (EM, Dempster et al. 1977) is a widely used paradigm to perform statistical estimation in latent variable models. The goal of EM is to maximise the *Free Energy*, a lower bound on the log-likelihood $\log p(\mathbf{s}; \mathbf{f})$ of the data, given by (following the notations of Section 2.1):

$$\mathcal{F}(\mathbf{f}, q) := \mathbb{E}_{q(\mathbf{x})}[\log p(\mathbf{x}, \mathbf{s}; \mathbf{f})] - \mathbb{E}_{q(\mathbf{x})}[\log q(\mathbf{x})] \leq \log p(\mathbf{s}; \mathbf{f}),$$

where $q$ is some probability distribution on the latent variable $\mathbf{x}$. Importantly $\mathcal{F}$ is maximised, and the lower bound becomes "tight", at $q^\star := p(\mathbf{x}|\mathbf{s}; \mathbf{f})$, i.e. the posterior distribution of the latent variable given the $\mathbf{s}$ and $\mathbf{f}$. Moreover, for a fixed $q$, the only $\mathbf{f}$-dependent term in $\mathcal{F}$ is $\mathbb{E}_{q(\mathbf{x})}[\log p(\mathbf{x}, \mathbf{s}; \mathbf{f})]$. To maximise $\mathcal{F}(\mathbf{f}, q)$ — and thus also increase the log-likelihood — EM produces a sequence $(\mathbf{f}^{(e)})_{e \geq 0}$ of parameters $\mathbf{f}^{(e)}$ by invoking, at each step (or "epoch") $e$, two well known subroutines:

- **E-step**: Define $q^{(e)} := p(\mathbf{x}|\mathbf{s}; \mathbf{f}^{(e-1)})$; compute $\mathcal{F} \longmapsto \mathbb{E}_{q^{(e)}}[\log p(\mathbf{x}, \mathbf{s}; \mathbf{f})]$
- **M-step**: Compute $\mathbf{f}^{(e)} := \arg\max_{\mathbf{f}} \mathcal{F}(\mathbf{f}, q^{(e)}) = \arg\max_{\mathbf{f}} \mathbb{E}_{q^{(e)}}[\log p(\mathbf{x}, \mathbf{s}; \mathbf{f})]$

with the property that $\log p(\mathbf{s}; \mathbf{f}^{(e)}) \geq \log p(\mathbf{s}; \mathbf{f}^{(e-1)})$ for all $e$, grounding the use of EM to maximise the likelihood of the data. In our context it is important to note that, due to a Gaussianity assumption, calculating the expectation in the E-step requires estimating the mean (and variance) of the posterior $p(\mathbf{x}|\mathbf{s}; \mathbf{f}^{(e-1)})$ which can be treated as a point estimate of the latent trajectory, i.e. a "decoding" of the latent from the spikes. Thus, in the context of neural data, EM offers a framework to both estimate intensity functions via maximum likelihood, and also to decode the variable encoded by the neurons.

**Impossibility of Exact EM for Gaussian-Modulated Poisson Processes** The $E$-step of the EM algorithm requires computing a function *defined* as an expectation w.r.t $p(\mathbf{x}|\mathbf{s}; \mathbf{f}^{(e-1)})$. In the case of Hidden Markov Models, such expectations are intractable to compute in closed form, unless the latent variable $\mathbf{x}$ is discrete (i.e. numerical estimation), or both the transition and the emission probabilities are Gaussian (with mean and variance depending linearly on $\mathbf{x}$, (Rauch et al., 1965)). In our particular case, exact inference in the model described in Section 2.1 is impossible because the emission probabilities are Poisson with mean given by a non-linear function of $\mathbf{x}$ via each neurons tuning curve.

In order to perform statistical inference for our spike train model — and avoid resorting to numerical estimation which is computationally expensive — SIMPL makes a set of approximations which we detail below. At a high level the goal is to convert the non-linear, non-Gaussian spiking observations, into a variable which is linear and Gaussian with resepct to the latent, thus EM can be performed exactly using a Kalman smoother.

### A.2  LINEAR GAUSSIAN STATE SPACE MODELS AND KALMAN SMOOTHING

Linear Gaussian State Space Models (LGSSM) are dynamical systems of the form:

$$\begin{aligned}
\mathbf{z}_{t+1} &= F_t \mathbf{z}_t + \epsilon_t, \quad \epsilon_t \sim \mathcal{N}(0_d, Q_t) \\
\mathbf{x}_t &= H_t \mathbf{z}_t + \delta_t, \quad \delta_t \sim \mathcal{N}(0_m, R_t).
\end{aligned} \tag{5}$$

where $\mathbf{z} \in \mathbb{R}^d$, $\mathbf{x} \in \mathbb{R}^m$, $F_t, Q_t \in \mathbb{R}^{d \times d}$, $H_t \in \mathbb{R}^{p \times d}$ and $R_t \in \mathbb{R}^{m \times m}$. LGSSMs can be used as latent variable models given some observed data $\mathbf{x}$, where $\mathbf{z}$ is treated as a latent variable. While these models are limited in their expressiveness, their benefits are that inference (here, the "E-steps") can be done very efficiently: not only is the posterior $p(\mathbf{z}_1, \ldots, \mathbf{z}_T | \mathbf{x}_1, \ldots, \mathbf{x}_T)$ a Gaussian distribution (of dimension $Td$), but all of its marginals and pairwise marginals

$p(\mathbf{z}_t|\mathbf{x}_1, \ldots, \mathbf{x}_T), p(\mathbf{z}_t, \mathbf{z}_{t+1}|\mathbf{x}_1, \ldots, \mathbf{x}_T)$ (crucially, the only distributions needed for learning the parameters of LGSSM via EM ) can be computed jointly in $\mathcal{O}(T)$ time using an efficient technique known as Kalman Smoothing (Kalman, 1960; Rauch et al., 1965).

Such a scaling contrasts with naive numerical binning-based alternatives for inference in continuous, non-Gaussian State Space Models, which require maintaining an estimate of each bin — a vector of size $n$ (no. bins) where $n$ grows *exponentially* with the dimension of the latent space, as used in e.g. Denovellis et al. 2021. Instead, for LGSSMs, the Gaussianity means only the mean and covariance of the marginal posterior distributions — of size $d$ and $d^2$ respectively — need to be stored. This is not memory intensive and, perhaps more importantly, the Kalman Filter proceeds to compute them in a combined $\mathcal{O}(T)$ time. In our experiments, we found the cost of the Kalman Filter to be negligible relative to the kernel evaluations which are the main computational bottleneck of SIMPL.

Note from here onwards we will return to using $\mathbf{x}$ to denote the latent variable in the LGSSM, and $\widehat{\mathbf{x}}$ or $\mathbf{s}$ for observations.

## B  SIMPL AS AN APPROXIMATE EM ALGORITHM

### B.1  MLE-BASED APPROXIMATE E-STEP

Instead of $q^{(e)} = p(\mathbf{x}|\mathbf{s}; \mathbf{f}^{(e-1)})$, SIMPL computes an approximation to $q^{(e)} \approx \widehat{q}^{(e)} = p(\mathbf{x}|\widehat{\mathbf{x}}; \mathbf{f}^{(e-1)})$ where $\widehat{\mathbf{x}}$ is the Maximum Likelihood Estimate (MLE) of $\mathbf{x}$ given the observations $\mathbf{s}$ and the current tuning curves $\mathbf{f}^{(e-1)}$ defined as:

$$\widehat{\mathbf{x}} = \arg\max_{\mathbf{x}} \log p(\mathbf{s}|\mathbf{x}; \mathbf{f}^{(e-1)}) = \arg\max_{\mathbf{x}} \sum_{t=1}^{T} \sum_{i=1}^{N} \log p(s_{ti}|\mathbf{x}_t; \mathbf{f}^{(e-1)})$$

$$\implies \widehat{\mathbf{x}}_t = \arg\max_{\mathbf{x}_t} \sum_{i=1}^{N} \log p(s_{ti}|\mathbf{x}_t; \mathbf{f}^{(e-1)}).$$

As defined, computing the MLE returns a point estimate of the *true* trajectory that led to the observed spike train $\mathbf{s}$, however we seek a posterior. In particular, MLE does not use the prior knowledge encoded by $p(\mathbf{x})$.

To find the approximate posterior we note that, as a function of $\mathbf{s}$, $\widehat{\mathbf{x}}$ is itself a random variable. In the many neurons limit, and under certain regularity assumptions, the distribution of this random variable converges to a Gaussian, a fact known as *asymptotic normality*. In other words; though $\mathbf{s}$ (conditioned on $\mathbf{x}$) is a non-Gaussian random variable, $\widehat{\mathbf{x}}$ (a deterministic function of $\mathbf{s}$) is approximately Gaussian in the many neurons limit and thus satisfied the conditions of the LGSSM.

We restate a formal statement of this asymptotic normality result in the case of independent, but non-identically distributed observations [3] originally established in Bradley & Gart (1962), and reformulated using the notations of the model at hand. For simplicity, we will consider the case where only $P$ distinct intensity functions $\mathbf{f}_1, \ldots, \mathbf{f}_P$ exist, although versions of this result exist without this assumption.

**Theorem B.1** (Asymptotic Normality of the MLE ). *Let* $\mathbf{x}_t^\star \in \mathbb{R}^d$. *Let* $\mathbf{s} = (s_{1t}, \ldots, s_{Nt})$ *be independent random variables with probability densities* $p(s_{ti}|\mathbf{x}_t^\star; \mathbf{f}_{t(i)})$, *where* $t(i) \in \{1, \ldots, P\}$ *is the index of the intensity function* $f_{t(i)}$ *that generated the spike train* $s_{ti}$. *For* $p \in 1, \ldots, P$, *denote* $n_p$ *the number of times the intensity function* $f_p$ *appeared in the sequence* $\mathbf{f}_{t(i)}$. *Assume that the MLE* $\widehat{\mathbf{x}}_t$ *exists and it is unique. Then, under mild regularity conditions, we have:*

$$\sqrt{N}\left(\widehat{\mathbf{x}}_t - \mathbf{x}_t^\star\right) \xrightarrow[N \to \infty]{\mathrm{d}} \mathcal{N}(0, \mathcal{I}(\mathbf{x}_t^\star)^{-1})$$

*where* $\mathcal{I}(\mathbf{x}_t^\star) := \sum_{p=1}^{P} \mu_p \mathbb{E}_{p(\mathbf{s}_t; \mathbf{f}_p)} \mathrm{Hess}(\log p(\mathbf{s}_t|\mathbf{x}_t^\star; \mathbf{f}_p))$ *is the Fisher Information matrix of the model at* $\mathbf{x}_t^\star$, $\xrightarrow{\mathrm{d}}$ *means convergence in distribution, and we defined* $\mu_p := \lim_{N \to \infty} \frac{n_p}{N}$.

---

[3] The i.i.d case was established in Fisher (1925)

The asymptotic Gaussianity of the MLE in the many neurons limit suggests performing approximate inference in a surrogate Hidden Markov Model, with the same transition probabilities $p(\mathbf{x}_{t+1}|\mathbf{x}_t)$ as the original ones, but where the observations $\mathbf{s}$ are replaced by $\widehat{\mathbf{x}}$. Leveraging Theorem B.1, SIMPL approximates the emission probabilities $p(\widehat{\mathbf{x}}_t|\mathbf{x}_t)$ by the Gaussian distribution $\mathcal{N}(\mathbf{x}_t, \boldsymbol{\Sigma}_t)$, where $\boldsymbol{\Sigma}_t := (N\mathcal{I}(\widehat{\mathbf{x}}_t))^{-1} \approx (N\mathcal{I}(\mathbf{x}_t))^{-1}$, the Fisher information of the spikes.

Temporarily ignoring the $\mathbf{x}_t$-dependence of the covariance matrices $\boldsymbol{\Sigma}_t$ and treating them as deterministic (discussed below), the variables $(\mathbf{x}_t, \widehat{\mathbf{x}}_t)$ then form the following latent variable system with hidden variables $\mathbf{x}_t$ and observed variables $\widehat{\mathbf{x}}_t$ given by:

$$
\begin{aligned}
\mathbf{x}_{t+1} \mid \mathbf{x}_t &\sim \mathcal{N}(\mathbf{x}_t, \sigma_v^2 \mathbf{I}), \\
\widehat{\mathbf{x}}_t \mid \mathbf{x}_t &\sim \mathcal{N}(\mathbf{x}_t, \boldsymbol{\Sigma}_t)
\end{aligned}
\tag{6}
$$

This model is precisely an instance of Linear Gaussian State Space Models defined in Equation 5 and the four matrices set to:

$$
\begin{aligned}
F_t &= \mathbf{I} && \text{(constant)} \\
H_t &= \mathbf{I} && \text{(constant)} \\
Q_t &= \sigma_v^2 \mathbf{I} && \text{(constant)} \\
R &= \boldsymbol{\Sigma}_t && \text{(time-varying)}.
\end{aligned}
$$

This correspondence allows SIMPL to compute an approximation of the marginal posterior distributions $p(\mathbf{x}_t|\mathbf{s}) \approx p(\mathbf{x}_t|\widehat{\mathbf{x}})$ using Kalman Smoothing (Kalman, 1960; Rauch et al., 1965). Importantly, the MLE estimates $\widehat{\mathbf{x}}_t$ can be obtained in parallel for all $t$; the only sequential procedure remaining being the Kalman Smoothing step. The trajectory $\widehat{\mathbf{x}}^{(e)}$ of SIMPL's E-step is then set to the mean of $\widehat{q}^{(e)}$.

### B.2 Spike Smoothing as a generalized M-Step

SIMPL's M-step is computationally cheap and interpretable. However, it differs from the M step of the EM algorithm, which we recall is given by:

$$
\mathbf{f}^{(e),\text{EM}} := \arg\max_{\mathbf{f}} \mathcal{F}(\mathbf{f}, q^{(e)}) = \arg\max_{\mathbf{f}} \mathbb{E}_{q^{(e)}}[\log p(\mathbf{x}, \mathbf{s}; \mathbf{f})]
\tag{7}
$$

Below, we reconcile the two approaches by showing that SIMPL M-step can be seen as instances of a more general "model fitting" M-step.

To see why, note the objective function of a standard M-step equals (up to a constant in $\mathbf{f}$) the negative KL divergence between the joint distribution [4] $\widehat{q}^{(e)}(\mathbf{x}|\mathbf{s})p(\mathbf{s})$ (observable through samples) and the model $p(\mathbf{s}, \mathbf{x}; \mathbf{f})$. Thus, a standard M-step can be understood as minimizing this KL divergence approximately, by replacing the expectation over $p(\mathbf{s})$ by an empirical average over the true data $\mathbf{s}$, an approximation which is asymptotically consistent in the large number of time steps limit under suitable ergodicity conditions (Billingsley, 1961).

Similarly, SIMPL's M-Step also fits the model $p(\mathbf{s}, \mathbf{x}, \mathbf{f})$ to the "data" distribution $\widehat{q}^{(e)}(\mathbf{x}|\mathbf{s})p(\mathbf{s})$. However, instead of doing so by minimizing the KL divergence between the data and the model, it does so using a kernel-based estimate [5]. Thus, Both M-steps can be understood as having the same goal, simply differing in their solution to solve it. In that sense, SIMPL's M-step is indeed a generalized M-step.

## C Implementation Details

Below we provide some implementation details that were important to maximise the computational efficiency of the method.

---

[4] We denote $q^k(x)$ by $q^k(x|s)$ to highlight the dependence between $x$ and $s$.

[5] Additionally, it replaces the expectation over $\widehat{q}^{(e)}(\mathbf{x}|\mathbf{s})$ by a one-sample estimate of it through $\tilde{\mathbf{x}}$

## C.1 MAXIMIZING SIMPL'S COMPUTATIONAL EFFICIENCY

### C.1.1 COMPUTATIONAL BOTTLENECKS IN SIMPL

A single evaluation of the log-likelihood $\log p(\mathbf{s}_t|\mathbf{x}_t)$ requires evaluating the kernel-based rate map estimates given in Equation 7. This takes $\mathcal{O}(T)$ time since it involves a sum across all timesteps. Moreover, this calculation will be repeated itself $T$-times for each step of the Kalman smoother in order to (1) compute the MLEs $\widehat{\mathbf{x}}_t$ (which naively require gradient ascent on $\log p(\mathbf{s}_t|\mathbf{x}_t)$) and (2) evaluate the MLE variance $\boldsymbol{\Sigma}_t := (N\mathcal{I}(\widehat{\mathbf{x}}_t))^{-1} = (N\mathbf{H}_x(\log p(\mathbf{s}|\widehat{\mathbf{x}}_t))(\widehat{\mathbf{x}}_t))^{-1}$. All in all, an exact implementation of SIMPL's E-step would have quadratic $\mathcal{O}(T^2)$ time complexity, which would be prohibitively slow for long datasets. Moreover, the second-order differentiation needed to compute $\mathcal{I}(\widehat{\mathbf{x}}_t)$ is also computationally expensive (formally, in introduces a large constant factor in front of the $O(T^2)$ term).

In the next sections, we describe additional approximations which allow SIMPL to estimate the MLE and its variance in $\mathcal{O}(T)$ time and without differentiating the rate maps.

### C.1.2 LINEAR-TIME MLE ESTIMATION

**Naive gradient-based solution** The naive way to calculate the MLE $\widehat{\mathbf{x}}_t$ is to evaluate all $N$ tuning curves (recall each evaluation costs $\mathcal{O}(T)$) for some location $\mathbf{x}$, use these to establish the log-likelihood $\log p(\mathbf{s}_t|\mathbf{x})$, calculate the gradient of this log-likelihood w.r.t. $\mathbf{x}$, and then take, for example, $k$ gradient descent steps to find the MLE. This process is repeated for each timestep $t$ in the Kalman smoother, leading to a quadratic time complexity of $\mathcal{O}(kNT^2)$.

**SIMPL's approach** To compute the MLE in linear time SIMPL bypasses the need to recalculate the tuning curves at each time step by, instead, binning them onto a discretised grid of points once at the start of each iteration.

Formally SIMPL computes $n$ evaluations the tuning curves $\tilde{\mathbf{f}} := (\tilde{\mathbf{f}}_1, \dots, \tilde{\mathbf{f}}_n) := (\mathbf{f}(\mathbf{g}_1), \dots, \mathbf{f}(\mathbf{g}_n))$ on a grid of $n$ points $\mathcal{G} = (\mathbf{g}_1, \dots, \mathbf{g}_n)$. This has time complexity $\mathcal{O}(NnT)$. We use a uniform rectangular grid of points (the smallest rectangle containing the full observed behavioural variable) of spacing $dx$. For example, in a 1 m × 1 m environment with $dx = 0.02$ m, this would yield a grid of 50×50 points ($n = 2500$).

Then, given $\tilde{\mathbf{f}}$, SIMPL then discretizes the log-likelihood functions $\log p(\mathbf{s}_t|\mathbf{x})$ over that same grid:

$$\tilde{l}_{it} := \log p(\mathbf{s}_t|\mathbf{g}_i) = \sum_{j=1}^{N} \log p(s_{tj}|\mathbf{g}_i) = \sum_{j=1}^{N} \log \frac{e^{-\tilde{f}_{ij}} \tilde{f}_{ij}^{s_{tj}}}{s_{tj}!}$$
$$= -\sum_{j=1}^{N} \tilde{f}_{ij} + s_{tj} \log \tilde{f}_{ij} - \log s_{tj}!$$

(8)

where we noted $\tilde{f}_{ij} := [\tilde{\mathbf{f}}(\mathbf{g}_i)]_j$. Finally, given such evaluations, SIMPL sets its approximation of the MLE to be

$$\widehat{\mathbf{x}}_t := \arg \max_{\mathbf{g} \in \mathcal{G}} \log p(\mathbf{s}_t|\mathbf{g}) = \arg \max_i \tilde{l}_{it}$$

This way of calculating the MLE has linear time complexity yielding an improvement for $n < kT$.

### C.1.3 LINEAR-TIME DERIVATIVE-FREE MLE VARIANCE ESTIMATION

A similar strategy could be employed to also compute $\mathcal{I}(\widehat{\mathbf{x}}_t) := -\mathbf{H}_x(\log p(\mathbf{s}_t|\widehat{\mathbf{x}}_t))(\widehat{\mathbf{x}}_t)$, which appears in $\boldsymbol{\Sigma}_t$. Here $\mathbf{H}_x$ is the Hessian operator defined as $\mathbf{H}_x(f)(x) := \nabla_x^2 f(x)$. To do so, one could compute the Hessian of the rate maps and their logarithm on that grid, from which any $\mathbf{H}_x(\log p(\mathbf{s}|\widehat{\mathbf{x}}_t))(\widehat{\mathbf{x}}_t)$ at the grid-point-based MLE obtained above can be evaluated as $\mathbf{H}_x(\log p(\mathbf{s}_t|\mathbf{g}_i))(\mathbf{g}_i) = -\sum_{j=1}^{N} \mathbf{H}_x(f_j)(\mathbf{g}_i) + s_{tj}\mathbf{H}_x(\log f_j)(\mathbf{g}_i)$. This would be linear in time however, we found that differentiating $\mathbf{f}$ could be very slow.

Instead SIMPL takes an entirely different approach and produces an estimation of $\boldsymbol{\Sigma}_t$ by instead *estimating the variance of the posterior distribution* $p(\mathbf{x}_t|\mathbf{s}_t) \propto p(\mathbf{x}_t)p(\mathbf{s}_t|\mathbf{x}_t) = p(\mathbf{s}_t, \mathbf{x}_t)$. The

posterior variance and the MLE variance are expected to closely match, as discussed in our theoretical justification above. Moreover, as this posterior is available analytically up to the normalizing constant $p(\mathbf{s}_t)$, its variance can be approximately computed by binning $p(\mathbf{x}_t|\mathbf{s}_t)$ onto the same grid $\mathcal{G}$ introduced above, yielding the following fast estimator for $\boldsymbol{\Sigma}_t$.

$$\boldsymbol{\Sigma}_t \approx \text{Cov}\, p(\mathbf{x}_t|\mathbf{s}_t) \approx \frac{\sum_i \tilde{p}_{it}(\mathbf{g}_i - \boldsymbol{\mu}_t)(\mathbf{g}_i - \boldsymbol{\mu}_t)^T}{\sum_i \tilde{p}_{it}}, \quad \boldsymbol{\mu}_t := \frac{\sum_i \mathbf{g}_i \tilde{p}_{it}}{\sum_i \tilde{p}_{it}} \tag{9}$$

where $\tilde{p}_{it} := \exp(\tilde{l}_{it}) = p(\mathbf{s}_t|\mathbf{g}_i)$. Intuitively, this is equivalent to fitting a multivariate Gaussian to the binned likelihood map. The covariance matrix of this Gaussian is then used as an approximation of the MLE variance. We provide a theoretical argument justifying the validity of this formula below.

**Theoretical Justification**   Equation 9 is justified by the Bernstein Von Mises theorem, which states that the difference between the posterior distribution and the distribution of the MLE vanishes in the many neurons limit. We restate this theorem using the notations of our paper, assuming a unique rate map, and without stating some of the required regularity assumptions for simplicity. We refer the reader to (Van der Vaart, 2000, Theorem 10.1, p.141–144) for the full version.

**Theorem C.1** (Bernstein-von Mises). *Let $\mathbf{x}_t^\star \in \mathbb{R}^d$. Let $\mathbf{s}_t = (s_{1t}, \ldots, s_{Nt})$ be i.i.d random variables with probability density $p(\mathbf{s}_t|\mathbf{x}_t^\star; \mathbf{f})$. Assume that the MLE $\widehat{\mathbf{x}}_t$ exists and it is unique. Then, under mild regularity conditions, for any prior $p$ on $\mathbf{x}_t$, we have:*

$$\|p(\mathbf{x}_t|\mathbf{s}_t) - \mathcal{N}(\widehat{\mathbf{x}}_t, (N\mathcal{I}(\mathbf{x}_t^\star))^{-1})\|_{\text{TV}} \underset{N\to\infty}{\overset{p(\mathbf{s}_t)}{\rightarrow}} 0$$

*where $\overset{p(\mathbf{s})}{\rightarrow}$ denotes convergence in probability, and $\|\cdot\|_{\text{TV}}$ denotes the Total Variation norm on bounded measures.*

From this theorem, we thus have that the (random) posterior distribution behaves (in total variation) as a Gaussian whose covariance matrix is precisely the asymptotic variance of the MLE. Note however that convergence in total variation does not a priori imply convergence of variances. Further work could examine under which assumptions such a convergence of variances may hold. In practice, we found that this approximation yielded a satisfying trade-off between performance and accuracy.

## C.2   ITERATIVE LINEAR REALIGNMENT OF THE TRAJECTORIES

To improve the identifiability properties and the numerical stability of SIMPL, we also transform the decoded latent trajectory at each iteration using a linear mapping which maximally aligns it with behaviour defined as $\mathbf{x}_t^{(e)} \leftarrow \mathbf{M}\mathbf{x}_t^{(e)} + \mathbf{c}$ where $\mathbf{M}, \mathbf{c} = \arg\min \sum_t \|\mathbf{x}_t^{(0)} - (\mathbf{M}\mathbf{x}_t^{(e)} + \mathbf{c})\|$. This approach ensures the scale, orientation and centre of the optimised latent trajectory are tied to behaviour, preventing accumulation of linear shifts/rotations across iterations and allowing us to interpret the latent relative to, and in the same units as, behaviour. We suspect that performing this alignment on all iterates *after* the optimisation would yield similar results. Because the transformed latent necessarily has similar scale to the behaviour — which was used to set the size of the discretised environment — we can reuse the same discrete grid for the latent avoiding the need to rediscretize the environment at each iteration.

## C.3   HYPERPARAMETERS SETTINGS

SIMPL has two model hyperparameters:

- $v$: the diffusion rate for Kalman smoothing, which sets a prior over expected velocity of the latent variable. Units are in $\text{ms}^{-1}$.
- $\sigma$: the bandwidth of the kernel used in the M-step to smooth spikes. Units are in m.

Additionally there are some implementation-specific parameters:

- $\mathrm{d}x$: the bin size for the variance estimation of the MLE. Units are in m.

- $\mathrm{d}t$: the time step of the discretization of the latent variable. Units are in s.

- $E$: the number of iterations of the EM algorithm.

Finally, in all simulations we used a test fraction of 10% and held out 'speckled' data segments of length 1 second to evaluate the performance of the model. We provide in Table 1 the value of these hyperparameters for the Artificial Grid Cell Dataset and the Real Hippocampal Dataset.

Table 1: Hyperparameters settings

| **Dataset** | $v$ | $\sigma$ | $\mathrm{d}x$ | $\mathrm{d}t$ | $E$ |
|---|---|---|---|---|---|
| Artificial Grid Cell Dataset (Fig. 2) | 0.4 ms$^{-1}$ | 0.02 m | 0.02 m | 0.1 s | 10 |
| Real Hippocampal Dataset (Fig. 3) | 1.0 ms$^{-1}$ | 0.1 m | 0.04 m | 0.2 s | 10 |
| Motor task dataset (2D) [6] (Fig. 4c&d) | 1.0 | 0.1 | 0.02 | 0.05 s | 10 |
| Motor task dataset (4D) (Fig. 4e) | 1.5 | 0.09 | 0.1 | 0.05 s | 10 |

## C.4 SYNTHETIC DATA GENERATION WITH THE RATINABOX PACKAGE

All synthetic grid cell data were generated using the RatInABox package (George et al., 2024a). In this model, an agent moves through a 1 m by 1 m environment following a smooth, continuous random motion policy (details can be found in the original paper)under default parameters whereby the agent's mean speed is 0.08 m/s. Specifically, the RatInABox model was used to generate the true latent trajectory, denoted as $\mathbf{x}^\star$. This trajectory was then "noised" to produce the trajectory used as the initial condition for SIMPL, denoted as $\mathbf{x}^{(0)}$, which represents the animal's measured position. The noise, or "discrepancy" vector $\Delta\mathbf{x} := \mathbf{x}^{(0)} - \mathbf{x}^\star$, was generated by sampling a velocity trajectory from a 2D Ornstein-Uhlenbeck process with zero mean and a coherence time scale of 3 seconds. This velocity trajectory was then integrated to obtain the discrepancy. Finally, the agent's behaviour $\mathbf{x}^{(0)}$ was additionally influenced by the same environmental forces implemented in the standard RatInABox model, i.e. the agent smoothly drifts away from walls to avoid crashing. The scale of the Ornstein-Uhlenbeck process was adjusted so that the mean discrepancy between the latent and observed trajectories was 20 cm in pen-space (i.e. away from walls).

Grid cells were modelled as the thresholded sum of three cosine plane waves (see RatInABox methods) with a width ratio—ratio of field width to inter-field distance—of 0.55. Each grid cell is assigned a wave direction $\theta_i$, gridscale $\lambda_i$ and 2D phase offset $\phi_i = [\phi_1, \phi_2]^\mathsf{T}$. Specifically, N=225 grid cells were divided into 3 modules of N=75 grid cells. Within each module all cells had the same grid scale (0.3, 0.5 and 0.8 m) and wave direction (0, 0.1 and 0.2 rad) but random phase offsets, approximately matching grid cells in the brain. Grid cell firing rates are all scaled to a maximum of 10 Hz.

Grid cells firing rates are determined by the latent position at time $t$, i.e. $f_i(t) = f^{GC}(\mathbf{x}_t^\star)$, from which spikes are sampled according to an inhomogeneous Poisson process $N_{spikes}(t, t + dt) \sim \mathrm{Poi}(f_i(t) \cdot dt)$

## C.5 TEST-TRAIN PARTITIONING

To assess performance we partition the spike data matrix, $\mathbf{s}$, into testing and training sets, $\mathcal{S}_{\text{test}}, \mathcal{S}_{\text{train}}$. Inference is performed solely on the training set and we then track the log-likelihood of data in both sets (Fig. 2d, left), e.g. $\ell^{(e)} = |\mathcal{S}_{\text{test}}|_{\text{test}}^{-1} \sum_{(i,t) \sim \mathcal{S}^{\text{test}}} \log p(s_{ti}|\mathbf{x}_t^{(e)}, \mathbf{f}_i^{(e)})$. This partitioning requires careful consideration: entire time intervals cannot be withheld for testing without impairing the model's ability to infer the latent over this period. Likewise, entire neurons cannot be withheld without impairing the model's capacity to estimate their tuning curves. Instead, we adopt a speckled train-test mask previously used in latent variable modelling set-ups (Williams et al., 2020) which withholds for testing extended chunks of time bins arranged in an irregular "speckled" pattern across the data matrix (totalling 10% of the data).

## C.6 BENCHMARKING DETAILS

In section 3.5, we benchmarked SIMPL against four comparable methods on the synthetic grid cell datasets. For all techniques, tuning curves were visualised in the same way as for SIMPL: by extracting the latent trajectories after optimisation and using kernel smoothing to construct the rate maps, i.e., a single "M-step." Default parameters were used for all methods. CEBRA constrains its N-dimensional latent space to lie on an (N-1)-dimensional hypersphere. We disabled this constraint for the 2D grid cell dataset. For GPLVM, we used a variant that exploits induction points to improve scalability with the amount of data, performed a grid-search across the number induction points, and reported the best result. For GPDM, which does not feature induction points and thus has a cubic complexity w.r.t $T$, we restricted training for this technique to 5 minutes of data (compared to 60 minutes for the other methods) to keep computation times under 1 hour,. For pi-VAE, we set the "task variable" $u_t$ to the behaviour $x_t^{(0)}$, and the latent $z_t$ to be the true position $x_t$. Additionally, we set the distribution $p(z_t|u_t)$ (e.g. the position given behavior mapping) to be a Gaussian distribution centered at $u_t = x_t^{(0)}$, and a fixed variance.

# D ADDITIONAL RESULTS

## D.1 TOY MODEL OF A DISCRETE LATENT VARIABLE TASK

Before testing SIMPL on a large temporally continuous dataset we constructed a smaller dataset akin to a discrete two-alternative forced choice task (2AFC) (Fig. 7) — a widely studied decision–making paradigm (Platt & Glimcher, 1999; Bogacz et al., 2006; Znamenskiy & Zador, 2013; Lieder et al., 2019). The true latent states $\mathbf{x}_t^\star \in \{0, 1\}$ are binary and have no temporal structure (here subscript $t$ indexes *trials* not time), analogous to a series of random "left" or "right" choices (Fig. 7b). This latent state is stochastically encoded by a population of neurons with random tuning curves giving the Bernoulli emission probabilities under each latent state:

$$f_i^\star(\mathbf{x}) = \begin{cases} f_{i0} \sim \mathcal{U}(0, 1) & \mathbf{x} = 0, \\ f_{i1} \sim \mathcal{U}(0, 1) & \mathbf{x} = 1, \end{cases}$$

$$\mathbf{x}_t^\star \sim \text{Bernoulli}(0.5) \quad \text{and} \quad s_{ti}|\mathbf{x}_t \sim \text{Bernoulli}(f_i^\star(\mathbf{x}_t^\star)).$$

Data is then sampled for $T = 50$ trials and $N = 15$ neurons as shown in Fig. 7. Initial conditions, $\mathbf{x}_t^{(0)}$, are generated from the true latent by randomly resampling a fraction of trials $\rho = 0.5$ (Fig. 7b). This partial resample represents an initial discrepancy between the behavioural measurement and the true internal state of the agent.

We perform inference on this dataset using a reduced version of the model (SIMPL-R). In the M-step, tuning curves were fitted by calculating the average activity of a neuron across each latent condition (e.g. $f_i^{(e)}(\mathbf{x}) = \sum_t s_{ti}\delta(\mathbf{x}_t^{(e)}, \mathbf{x})/\sum_t \delta(\mathbf{x}_t^{(e)}, \mathbf{x})$, conceptually similar to kernel smoothing). For the E-step, each latent was the decoded according to the maximum likelihood estimate under the observed spikes and tuning curve estimates from the previous epoch: $\mathbf{x}_t^{(e+1)} = \arg\max_\mathbf{x} \sum_i \log p(s_{ti}|\mathbf{x}, f_i^{(e)})$ (there is no time dependence between latents, thus no Kalman smoothing). This process was repeated for 5 epochs and, with high reliability, converged on the true latents after approximately two (Fig. 7c & d, distributions show repeat for 1000 randomly seeded datasets, dotted lines show ceiling performance on a model perfectly initialised with noiseless $\mathbf{x}^{(0)} = \mathbf{x}^\star$). We repeated this experiment for various values of $\rho$: latent recovery was almost perfect when $\rho$ was small (i.e. when the initial conditions were close to the true latent), dropping off as $\rho$ approached 1. At $\rho = 1$ when the conditions were *completely* random, the model was biased to recover a latent space that is either perfectly correlated or perfectly anti-correlated ("left" $\leftrightarrow$ "right") with the true latent (Fig. 7c, right), an isomorphic solution.

## D.2 HYPERPARAMETER SWEEP

We swept over the two hyperparameters $v$ (the velocity prior) and $\sigma$ (the kernel bandwidth) to assess how sensitive SIMPL is to these hyperparameters, as shown in Figure 8. For this we used the same synthetic grid cell dataset used in Fig. 2. Notably, SIMPL's performance (measured in terms

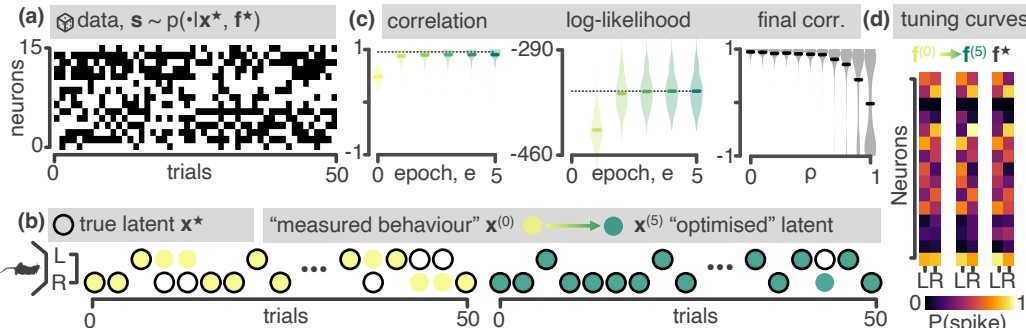

Figure 7: A two-alternative forced choice task (2AFC) toy-model. **(a)** Data generation: Spikes are sampled from a simple generative model. For each of T=50 independent trials a random binary latent — analogous to a "left" or "right" choice — is encoded by a population of N=15 neurons with randomly initialised tuning curves. **(b)** Model performance: Starting from a noisy estimate (yellow) of the true latent (black) where a fraction $\rho = 0.5$ of trials are resampled, SIMPL-R recovers the true latent variables (green) with high accuracy. **(c)** *Left:* Correlation between $\mathbf{x}^{(e)}$ and $\mathbf{x}^{\star}$. *Middle:* Log-likelihood, $\log p(\mathbf{s}|\mathbf{x}^{(e)}, \mathbf{f}^{(e)})$. *Right:* Final correlation between $\mathbf{x}^{(5)}$ and $\mathbf{x}^{\star}$ as a function of initialization noise $\rho$. Violin plots show distributions over 1000 randomly seeded datasets, dotted lines show ceiling performance of a perfectly initialised model ($\mathbf{x}^{(0)} = \mathbf{x}^{\star}$) **(d)** Tuning curves.

of the final error, see panel b) is relatively stable across a wide range of hyperparameters; kernel bandwidths between 0.1 cm and 5 cm and velocity priors between 0.2 m/s and 1 m/s all yield similar performance. When the tuning curves are confirmed that kernel bandwidth has a significant effect on their appearance. Broader kernels give smoother tuning curves eventually blurring the individual grid fields together whilst narrower kernels give sharper tuning curves eventual leading to overfitting where individual spikes are resolved.

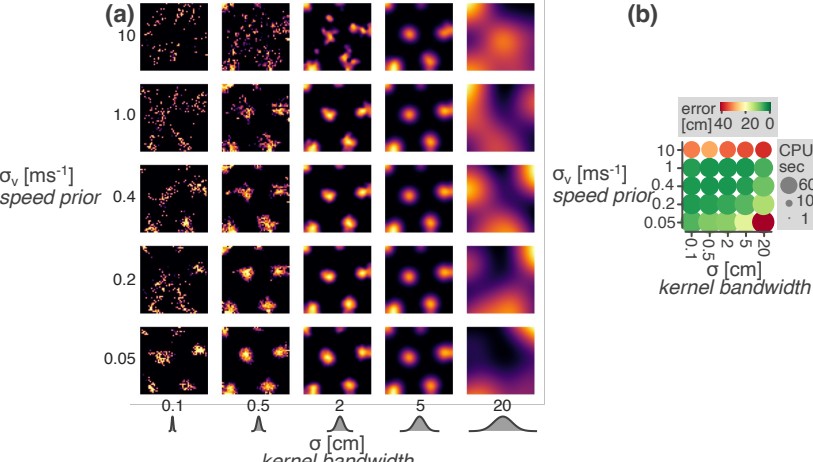

Figure 8: Performance of SIMPL on the synthetic grid cell dataset as a function of the hyperparameters $v$ (speed prior) and $\sigma$ (kernel bandwidth). **(a)** Tuning curves. **(b)** Final error between the latent and ground truth (colour) and total compute time (size).

### D.3 NON-CONTINUOUS HIPPOCAMPAL REPLAY DATASET

Since SIMPL places an explicit prior on latent trajectories which are smooth and continuous we tested whether it could be used to model a dataset where the latent variable is non-continuous. For this we simulated a synthetic "replay" dataset from $N = 225$ small Gaussian place cells. In this dataset the latent variable and behaviour perfectly match except for regular, brief periods of "replay" where the latent variable jumps to a new location. Using the same hyperparameters as in the main text we found that SIMPL was able to recover the latent variable, capturing (or "decoding") the replay events with high accuracy (Fig. 9), despite its smoothness prior.

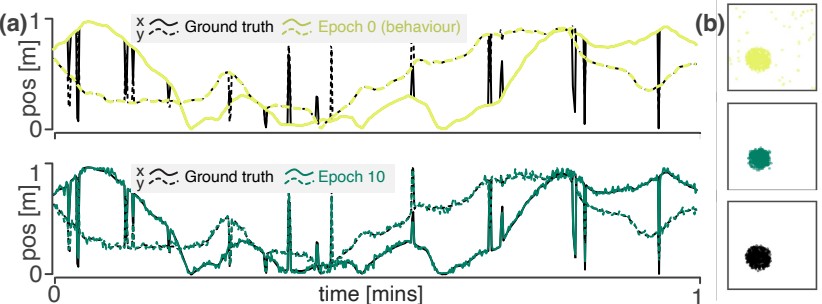

Figure 9: A synthetic hippocampal "replay" dataset. **(a)** One minute of trajectory, x-coordinate in solid line, y-coordinate in dashed. The behaviour (light-green, top panel) is smooth, actually matching the latent most of the time except when the latent takes regular, brief discontinuous jumps reminiscent of hippocampal replay events. After optimisation SIMPL is able to recover the latent (dark-green, bottom panel) and capture the replay events with high accuracy. **(b)** Spike raster plots; spikes plotted against the behaviour, optimised latent and ground truth latent.

## D.4 AUTOMATIC PLACE FIELD DETECTION

In Fig. 3, it was shown that the tuning curves of place cells in the hippocampus undergo statistically significant changes when optimised using SIMPL. For this analysis, individual place fields were automatically identified from the binned rate maps as isolated regions of elevated activity within a cells tuning curve. This was done by thresholding the activity of each neuron at 1 Hz and identifying contiguous regions of activity with a peak firing rate above 2 Hz and a total area less than half that of the full environment, similar to approaches taken in previous work (Tanni et al., 2022).

## E SUMMARY TABLE OF RELATED METHODS

Here we summarize some of the most relevant LVM and dimensionality reduction techniques in the context of our five key desiderata as described in the related work section. These are:

1. Complex tuning curves: Does the model learn/infer non-linear tuning curves as opposed to linear/exponential-linear/etc. tuning curves.

2. Smooth latent dynamics: Does the model impose smooth temporal dynamics on the latent space (e.g. by assuming a linear dynamical system, Gaussian process or using an RNN), as opposed to treating each time point independently.

3. Spike-friendly: Was the method designed for spiking data. For probabilistic models, this refers to whether the generative noise model is Poisson as opposed to, say, Gaussian.

4. Exploit behaviour: Does/can the model use behaviour (as an observation, contrastive loss-target, initialisation, or otherwise) to guide latent discovery.

5. Scalable: Can the model scale to datasets of long duration. Specifically, in available open-source implementations of the method does training/inference have near-linear time complexity. Note this does *not* mean that compute time is necessarily fast in an absolute sense, just that scaling is linear.

| Model | Complex tuning curves | Smooth latent dynamics | Spike-friendly | Exploit behaviour | Scalable |
|---|---|---|---|---|---|
| **SIMPL (Our method)** | Yes | Yes | Yes | Yes | Yes |
| **GPLVM (Lawrence, 2003)** | Yes | N/S | No | Yes[§] | Yes[#] |
| P-GPLVM (Wu et al., 2017) | Yes | Yes | Yes | Yes[§] | No |
| M-GPLVM (Jensen et al., 2020) | Yes | No | No | No | Yes[#] |
| faeLVM (Bjerke et al., 2023) | Yes[†] | Yes | Yes | No | Yes |
| PfLDS (Gao et al., 2016) | Yes | Yes | Yes | No | Yes |
| VIND (Hernandez et al., 2018) | Yes | Yes | Yes | No | Yes |
| **pi-VAE (Zhou & Wei, 2020)** | Yes | No | Yes | Yes | Yes |
| **CEBRA (Schneider et al., 2023)** | N/A | No | Yes | Yes | Yes |
| MIND(Low et al., 2018) | Yes | No | No | No | Yes |
| LFADS (Pandarinath et al., 2018) | No | Yes | Yes | Yes | Yes |
| TNDM (Hurwitz et al., 2021) | No | Yes | Yes | Yes | Yes |
| GP-SDEs (Duncker et al., 2019) | No | Yes | N/S | N/S | Yes |
| rSLDS (Linderman et al., 2016) | No | Yes | No | Yes[§] | Yes |
| gpSLDS (Hu et al., 2024) | No | Yes | No | Yes[§] | Yes |
| **GPDM (Wang et al., 2005)** | Yes | Yes | No | Yes[§] | No |
| MM-GPVAE (Gondur et al., 2023) | No | Yes | Yes | No | Yes |
| PSID (Sani et al., 2021) | No | Yes | No | Yes | Yes |
| GPFA (Yu et al., 2008a) | No | Yes | No | No | Yes[#] |
| P-GPFA (Nam, 2015) | No | Yes | Yes | No | No |
| SSMDM (Zoltowski et al., 2020) | No | Yes | Yes | Yes | Yes |
| PLNDE (Kim et al., 2021) | No | Yes | Yes | Yes | Yes |
| GLDS (Kalman, 1960) | No | Yes | No | No | Yes |
| DKF (Krishnan et al., 2015) | Yes | Yes | No | No | Yes |
| PLDS (Macke et al., 2011) | No | Yes | Yes | No | Yes |
| UMAP (McInnes et al., 2018) | N/A | No | No | No | No |
| TSNE (Van der Maaten & Hinton, 2008) | N/A | No | No | No | No |
| pPCA (Pearson, 1901; Tipping & Bishop, 1999) | No | No | No | No | Yes |
| dPCA (Kobak et al., 2016) | No | No | No | Yes | Yes |

Table 3: A table of comparable models and their properties. N/A means the criterion is not applicable to the model. N/S means the criterion is not specified or may be dependent on implementation specifics. Techniques in **bold** are compared to on our benchmark dataset in Fig. 6
[#]: Scalable if using an implementation making induction point approximations.
[§]: Algorithm could be initialised at behaviour.
[†]: Assumes neurons in a given ensemble have a shared tuning curve structure (e.g. Gaussian) with neuron-specific transformations (e.g. shift and scale)

