# OpenReview forum: "SIMPL: Scalable and hassle-free optimisation of neural representations from behaviour"
_ICLR.cc/2025/Conference — ICLR 2025 Poster_

### Official Review · Reviewer_5Qja · 2024-10-26

**Soundness:** 3
**Presentation:** 2
**Contribution:** 2
**Rating:** 6
**Confidence:** 4

**Summary:**

The paper proposes a new method, SIMPL for estimating neural representation regarding behaviors. The method is based on the Expectation and Maximization method where during E-step, a model produces estimates of latent trajectories using Kalman smoothing and during M-step, the model uses KDE for fitting the intensity function.

**Strengths:**

* The paper is well-written. Especially, the references in the introduction can be a good guide to readers who are not familiar with neuroscience, especially, the hippocampus and MEC although the authors simply listed 10 papers in the actually related work section.

* The figure is generally helpful in understanding the method and the results.

* The proposed method is a very simple E-M-based algorithm but it is more efficient and has better performance than CEBRA in the grid cell simulation with the RatsInABox simulator.

**Weaknesses:**

1. The author claims general but limited to specific regions of brain place cells and grid cells. I highly recommend authors compare with various datasets such as the macaque dataset and the mouse visual cortex datasets used in Schneider et al. (2023).

2. The authors misuse big-O notation. For example, the authors mentioned O(1 hour, 200 neurons, 10^6 spikes) ~ O(1 min) in L 120 and O(hours) in L219. This causes confusion, especially to computer scientists who have been trained in big-O notation for measuring time or space complexity. If the authors want to use big-O notations, please change them to match the function of neurons/spikes and so on. However, I believe, just changing to plain text might be better.

3. Except for Section 4.2, there are no comparisons with existing methods or ablation studies. Although the results appear promising, it remains challenging to discern whether this is due to the simplicity of the task or the efficacy of the proposed method. Moreover, I am not sure whether CEBRA is the only method to compare with.
The authors should provide more details on the training and testing splits. L305–306 and 416–417 lack information: are the splits based on spatial segmentation (using specific parts of the box for training) or trial separation?

4. Although experiments with Tanni et al. (2022) represent the main result in this paper, the authors scarcely describe the task and the significance of each plot (particularly in Figure 6c), simply referring back to the original paper.

5. The authors compare SIMPL with CEBRA, a machine learning-based method running on CPUs. If they provide a runtime comparison showing that SIMPL on a CPU is faster than CEBRA on a GPU, it would further support their efficiency claims.

6. Lines 477–479 are ambiguous as to whether the authors discuss place cell remapping or another concept. If remapping is the intended topic, I recommend explicitly stating this and including a citation to aid readers from the machine learning community who may be unfamiliar with this concept.

7. The writing can be improved a lot:

7.1 It is possible that the readers do not know what the “turning curve” is since it is used in neuroscience literature and ICLR is generally a machine learning community. It is good to define what it is in the introduction.

7.2 It would be better to move the Related Work section to Before Method (standard ML conference styles) or Discussion (some ICML styles). Currently, it disconnects methods and results.

7.3 Please choose between British English and American English. Currently, it is used together (e.g., In L477, optimization, behaviour).

7.4 Consider that readers may not be familiar with “tuning curve,” a term more common in neuroscience literature. Since ICLR has a general machine learning audience, defining it in the introduction would be helpful.

7.5 Moving the Related Work section to a position before the Method section (per standard ML conference styles) or to the Discussion (in line with some ICML styles) may improve flow, as it currently separates the methods and results sections.

7.6 Maintain consistency in English dialect, as the paper currently alternates between British and American English (e.g., "optimization" vs. "behaviour" in line 477).

7.7 In line 131, specify “Appendix A” instead of “Appendix,” even if there is only one section.
7.8 Ensure the font for v is consistent in lines 142 and 143.

7.9 Since space permits, it may be preferable to adjust the placement of Figure 5 so that it doesn’t span pages 7 and 8, avoiding the empty space near lines 399–405.

7.10 IN L816, 1, …, P => \{ 1, …, P \}.

7.11 Correct citation formatting:
* In line 116, “Tanni et al. (2022)” should use citep.
* In lines 486, 493, and 790, change citep to citet.
* In line 522, switch citet to citep.

7.12 It would also be beneficial to unify the reference formatting:
* Journals like Nature and Nature Neuroscience do not list volume numbers, while JMLR and most eLife papers do (except in lines 583–585). Additionally, both “elife” and “Elife” are used inconsistently, as are “Nature” and “nature.”
* Clarify “cite” with a red background in L608.
* While “ICLR” is abbreviated, other conferences like NeurIPS and ICML are listed in full; consider unifying this approach.

**Questions:**

* What the arrow of epoch 1 -> 10 in Figure 3 is different from others? Does it have a specific meaning?

* What does “data now shown in L466 mean? Does it mean that the authors intentionally did not include the results since it is insignificant? If so, please include it in the appendix.

---

> ### Author Response · Authors · 2024-11-20
> **Response: New analysis on Macaque dataset and additional benchmarks**
>
> We thank the review for their comments. In response we have added new benchmarks with alternative techniques and new analysis on their suggested macaque dataset. As such we believe we have satisfied all of their concerns. We now respond to their comments point-by-point:
>
> > **"The author claims general but limited to specific regions of brain place cells and grid cells. I highly recommend authors compare with various datasets such as the macaque dataset and the mouse visual cortex datasets used in Schneider et al. (2023)."**
>
> Following this review we have now tested SIMPL on one of the Neural Latents Benchmark datasets (specifically the [Area2_Bump data](https://neurallatents.github.io/datasets) from somatosensory cortex for a macaque doing a centre-out reaching task collected by Chowdhury and Miller), the same dataset as used in the Schneider et al. (2023) paper. We find SIMPL works well (see the revised manuscript for a more detailed discussion of the results). In the results we have now added a new section and figure detailing our findings briefly summarised as follows:
> We tested 3 versions of SIMPL on the hand-reaching data.
> - SIMPL2D(position): The latent is initialised with the monkeys x- and y-hand position.
> - SIMPL2D(velocity): The latent is initialised with the monekys $v_x$- and $v_y$-hand velocity.
> - SIMPL4D(postion&velocity): A 4D latent is initialised with $x$, $y$, $v_x$ and $v_y$.
> In all three models SIMPL optimises the latent variable (the test-log-likelihood improves), uncovering a smooth latent variable correlated to (but substantially different from) the behavioural initiates. Corresponding tuning curves revealed neurons with "hand-position-like" or "hand-velocity-like" selective receptive fields. The 4D version of SIMPL performed better than either 2D version, revealing disentangled latents with a higher overall log-likelihood than either of the 2D models. Our finding reveal:
> 1. SIMPL can be applied to the types of non-hippocampal datasets commonly used in the LVM literature.
> 2. Both position and velocity _seperately_ do a good job explaining the latent dynamic but position and velocity _combined_ is better.
> 3. In the optimised latent space neurons have localised receptive fields reminiscent of place cells in the hippocampus.
> 4. SIMPL can be extended beyond the 2-dimensional latent spaces initially tested.
>
> > **"The authors misuse big-O notation."**
>
> We'll correct this for the camera-ready version.
>
> > **"3. Except for Section 4.2, there are no comparisons with existing methods"**
>
> Thank you for this suggestion, which was echoed by reviewer gB66 (where you will find a more detailed discussion). In summary, we have added two more comparsion to well known techniques (piVAE and GPLVM) which are, arguably, more relevant. In both cases these models perform well on the synthetic grid cell dataset but still worse than SIMPL and with over 10x compute cost. We think these additional benchmarks substantially strengthen our claim that SIMPL outperforms the most relevant, popular alternatives.
>
> > **"Although experiments with Tanni et al. (2022) represent the main result in this paper, the authors scarcely describe the task and the significance of each plot (particularly in Figure 6c), simply referring back to the original paper."**
>
> We will add a section to the discussion providing more details of this task. Regtarding panel 6c this shows violin plots of summary statistics for the tuning curves before and after SIMPL is applied, showing how they have changed. We'll clarify this for the camera ready.
>
> > **"The authors compare SIMPL with CEBRA, a machine learning-based method running on CPUs. If they provide a runtime comparison showing that SIMPL on a CPU is faster than CEBRA on a GPU, it would further support their efficiency claims."**
>
> Thank you for this suggestion however we don't think this is an appropriate or fair comparison. Because the compute-heavy step in SIMPL (calculating the likelihood maps, see Appendix for why) is parallelizable, we would also expect huge speed up for SIMPL on a GPU. Since it is all written in JAX this should be readily acheivable. If the reviewer still thinks it worthwhile we can add SIMPL-CEBRA GPU-vs-GPU time comparison. On the other hand, we believe one of the core values of SIMPL is it's speed _on a CPU_. GPU usage is a barreir for some researcher and not everybody has access to to these compute resources so we would still like to focus attention on CPU compute times. We hope this argument makes sense.

---

> > ### Comment · Reviewer_5Qja · 2024-11-21
> >
> > I appreciate the authors for dealing with my concerns. I will read revisions and other reviews to ask further questions and adjust my rating.
> >
> > > "The authors compare SIMPL with CEBRA, a machine learning-based method running on CPUs. If they provide a runtime comparison showing that SIMPL on a CPU is faster than CEBRA on a GPU, it would further support their efficiency claims."
> >
> >
> > Thank you for the clarification. The comment was not an important concern so I do not think the authors must use GPU to measure time right now.
> >
> >
> >
> > > The author claims general but limited to specific regions of brain place cells and grid cells. I highly recommend authors compare with various datasets such as the macaque dataset and the mouse visual cortex datasets used in Schneider et al. (2023)."
> >
> > It would be nice if the authors compared with CEBRA result in the same figure (Figure 7 in the revision).
> >
> >
> >
> > **Regarding response for big-O, description of Figure 6c, thread “...(response continued)”**
> >
> > I highly recommend fixing it during the reviewing period as one of the advantages of ICLR is the authors can freely revise the manuscript. Regarding ‘data not shown’ in L466, It would be nice if you included it in the supplementary materials for thoroughness.
> >
> > **Others**
> >
> > I believe the page limit is also applied to the revision. Could you revise the manuscript so that it fits on 10 pages?

---

> > > ### Author Response · Authors · 2024-11-24
> > >
> > > Thank you for your prompt response.
> > >
> > > > **"Regarding response for big-O....I highly recommend fixing it during the reviewing period"**
> > >
> > > We've gone ahead and corrected all of the minor formatting issues you mentioned (e.g. the misuse of big-O notation, citation style, British/American spelling and referencing inconsistencies).
> > >
> > > > **"It would be nice if the authors compared with CEBRA result in the same figure (Figure 7 in the revision)."**
> > >
> > > We have added a panel to Fig. 7e showing the equivalent CEBRA embedding. This data was taken and adapted directly from the CEBRA paper. We think it is a fair comparison to the SIMPL data we are showing alongside it in panel e since:
> > > * Both SIMPL and CEBRA are trained with 4D latent spaces
> > > * Both SIMPL and CEBRA align and average the latent across active-trials for plotting (-100ms to 500ms from movement onset time)
> > > * Both SIMPL and CEBRA use behaviour (e.g. hand-position) to inform (SIMPL-->initialise, CEBRA-->contrastive loss labels) latent discovery.
> > > This panels shows how SIMPL generates latent variables which are comparable to CEBRA (a more established technique).
> > >
> > > > **"Regarding ‘data not shown’ in L466, It would be nice if you included it in the supplementary materials for thoroughness."**
> > >
> > > It is now shown in the Appendix (Fig. 10).
> > >
> > > > **"I believe the page limit is also applied to the revision."**
> > >
> > > We had initially checked and the ICLR instructions are ambiguous on whether the 10 page limit applies to revisions as opposed to just initial and camera-ready submissions - we apologise for running over! For now we have moved Fig. 7 (new motor-task dataset) to the Appendix making it 10 pages again. In the camera-ready version this result will be moved back to the main text and Fig. 2 (the discrete-latent task, which in our opinion is less important) will be pushed to the Appendix.

---

> > > > ### Comment · Reviewer_5Qja · 2024-11-25
> > > >
> > > > I appreciate the authors for their sincere response to the clarification. I raised my rating.

---

> > > > > ### Author Response · Authors · 2024-11-25
> > > > >
> > > > > We thank the reviewer again for their review and comments which have improved the manuscript.

---

> ### Author Response · Authors · 2024-11-20
> **...(response continued)**
>
> >**"Lines 477–479 are ambiguous as to whether the authors discuss place cell remapping or another concept. If remapping is the intended topic, I recommend explicitly stating this and including a citation to aid readers from the machine learning community who may be unfamiliar with this concept."**
>
> We apologise for the confusion. We were not refering to hippocampal remapping in lines 477-479 but were discussing how the trajectory makes a discontinuous "jump" to another position in latent space. We will try and re-word in the camera-ready to avoid confusion.
>
> > **"The writing can be improved a lot"**
>
> Thank you for this very thorough check! We will certainly fix all of these errors in time for the camera-ready version.
>
> > **"What the arrow of epoch 1 -> 10 in Figure 3 is different from others? Does it have a specific meaning?"**
>
> This just refers to the fact that epochs 2--9 are collapsed and not shown to reduce clutter.
>
> > **"What does “data now shown in L466 mean? Does it mean that the authors intentionally did not include the results since it is insignificant? If so, please include it in the appendix."**
>
> Yes, we did no show this result because it is insignificant and we wanted to keep only the most important results in the main figure to reduce clutter. We will add these statistics in for the camera ready.
>
> We thank the review for their time and thorough review. We would be very happy to hear about, and address, about any remaining major concerns they may have regarding technical contributions, soundness and impact. We remain commited to addressing any concerns that may prevent the reviewer from recommending acceptance via an updated score.

---

### Official Review · Reviewer_gB66 · 2024-11-04

**Soundness:** 1
**Presentation:** 1
**Contribution:** 1
**Rating:** 5
**Confidence:** 4

**Summary:**

The authors propose a new latent variable model for neuroscience that uses kernel density estimation to learn tuning curves from an unobserved latent variable for population spiking data. The model is fit using a closed form EM algorithm and is scalable to large datasets. The authors validate the model on simulated and real neural data.

**Strengths:**

The validation on synthetic and real neural data is a nice addition. The simplicity of the approach has potentially some appeal to experimentalists who want to avoid some of the more bespoke or complex latent variable models for the neural spiking data.

**Weaknesses:**

The authors provide a very brief overview a very broad field of latent variable models used in neuroscience to motivate their approach. They claim existing methods do not scale well, or have other shortcomings compared to SIMPL. To robustly demonstrate that this model indeed scales better than most or all of these existing approaches while retaining accurate latent and neural tuning identification would require far more comparisons, discussion and evaluation.

In particular, the authors model is motivated in a similar way to the GPLVM, which is discussed but not compared to. As the authors point out GP based models often do have issues with scalability. However,  there have been many steps toward improving the scalability of these models in recent years -- note that 1 and 2 uses inducing points, a well-known approach to improve scalability in GP  models. These models however have the same 'tuning curve' interpretations that the authors posit SIMPL has, and so it would be important to directly compare to them for both scalability and neural tuning identification.

1 "Manifold GPLVMs for discovering non-Euclidean latent structure in neural data"
2. Learning interpretable continuous-time models of latent stochastic dynamical systems

There is also no mention of switching linear dynamical systems models, which often scale better than GP based methods due admitting an EM based approach and utilizing forward-backward passing (see e.g. 3, 4, but there are many others).

3 Bayesian learning and inference in recurrent switching linear dynamical systems.
4. A general recurrent state space framework for modeling neural dynamics during decision-making.

Still there are others in this space that scale well and admit flexible non-linear latent characterizations. See for example 4, 5. and of course LFADS, as the authors discuss. The authors also mention Pi-VAE (and there are of course now many other VAE based models used in neuroscience), but they don't provide a principled comparison or discussion of these approaches.

4 Inferring Latent Dynamics Underlying Neural Population Activity via Neural Differential Equations
5  Collapsed amortized variational inference for switching nonlinear dynamical systems.

The choice of CEBRA as the only benchmark is not well motivated. CEBRA is a fundamentally different model it uses behavioral information for contrastive learning, and it was designed to visualize a behaviorally-informed latent space. In this sense it isn't an unsupervised model used purely on spiking data (like SIMPL as well as the ones cited above, among others) and it does not permit a tuning curve interpretation -- which the authors emphasize is an important component of SIMPL and existing in many of these other LVMs. Comparison to the GPLVM, as well as switching LDS, or other nonlinear dynamical models would be far more appropriate then CEBRA.

In it's current state, it is unclear exactly how much better SIMPL scales than any of the existing approaches, and how it's latent identification or tuning identification may compare to these many state-of-the-art methods.

**Questions:**

None

---

> ### Author Response · Authors · 2024-11-20
> **Response: Additional benchmarks have been added and discussion regarding which models are appropriate benchmarks**
>
> We thank the reviewer for their very thorough review and for pointing us to this additional literature on latent variable modelling. We will cite these works in the camera-ready version of the paper. We provide below some important clarifications regarding which methods constitute relevant alternatives to SIMPL, as well as describing additional benchmarks we have run in response to the reviewers valid concerns.
>
> From a modelling standpoint SIMPL has four key properties which we view as desiderata for comparable techniques:
> 1. Complex, non-linear tuning curves
> 2. Time dependence of the latent variable.
> 3. Poisson emission probabilities
> 4. Identifiability from behaviour
>
> By identifiability we mean whether there are any reassurances (theoretical or empirical) that, given behaviour, the model can recover the true latent up to some affine transformation. Many models simply do not admit any way to consume behavioural information.
>
> We see desiderata 1 as non-negotiable since we are primarily focused on modelling cells with complex tuning curves (e.g. grid cells). Any model with a substantially restrictive intensity functions (specifically linear-type, of the form $y_t \sim \textrm{NoiseModel}(f(\mathbf{M}\mathbf{z}_t+\mathbf{c}))$ where $f$ is a simple function e.g. identity, softplus, exponential etc.) can never interpretably account the sorts of datasets (e.g. grid cells) we consider here. With that in mind we carefully checked all the references that the reviewer pointed us to. None of them satisify all these desiderata and, in particular, five of them (2, 3, 4, 5 and 7 as numbered in the table below), including LFADs, have linear-type tuning curves and, as such, we consider them mis-specified. Unless we have misunderstood, or the reviewer has a strong counter-reason, these method simply could not model the types of tuning curves we are interested in optimising and therefore we do not see them as valid comparisons. We will make this point much clearer in an updated manuscript.
>
> The following table summarises all papers mentioned by the reviewer and their status with respect to our four desiderata:
>
>
> | Paper | Complex tuning curves | Time dependence | Poisson emissions | Identifiability |
> |-------|-------|----|-----|----|
> |1. Manifold GPLVMs for discovering non-Euclidean latent structure in neural data|Y|Y|N|Y|
> |2. Learning interpretable continuous-time models of latent stochastic dynamical systems |N|Y|N|Y|
> |3.Bayesian learning and inference in recurrent switching linear dynamical systems.|N|Y|N|Y|
> |4. A general recurrent state space framework for modeling neural dynamics during decision-making.|N|Y|Y|Y|
> |5. LFADS|N|Y|Y|Y|
> |6. Pi-VAE|Y|N|Y|Y|
> |7. Inferring Latent Dynamics Underlying Neural Population Activity via Neural Differential Equations |N|Y|Y|Y|
> |8. Collapsed amortized variational inference for switching nonlinear dynamical systems.|Y|Y|N (strictly categorical) |N|
> | 9. GPLVM | Y | Y | N | Y |
>
> To be clear, the fact none of these references satisfy all four desiderata does not mean that all these methods aren't useful in cases where SIMPL could be used however it means that we believe, independent of explicit comparisons, that SIMPL still represents a valuable contribution to the field.
>
> More broadly, the closest alternative to SIMPL is possibly Poisson-GPLVM (Wu et al. 2017). However, unfortunately, the current P-GPLVM algorithm doesn't make use of inducing points, resulting in cubic complexity. We tried running this on 1000 datapoints and got a compute time of ~2 mins, already exceeding the runtime of SIMPL on _all_ 36000 datapoints by a factor of ~3. We would be happy see future work directed at improving the scalability of P-GPLVM however, in this work, we have taken a different approach by defining a model that bypasses GPs altogether.
>
> Nonetheless, we take seriously the reviewers suggestion that we need to improve our comparisons of SIMPL to alternatives. For this reason we have added comparisons to two other methods which only lack one of the desiderata listed above. Pi-VAE which imposes _time independence_ on the latent (i.e. only fails desiderata 2) and GPLVM which assumes _Gaussian emissions_ (desiderata 3). Both come with well maintained code bases. The GPLVM algorithm can exploit inducing points to make it scalable. Results are shown in the updated version of the paper. In summary:
>
> (continued in next comment...)

---

> ### Author Response · Authors · 2024-11-20
> **...(response continued)**
>
> * GPLVM: We find that GPLVM performs quite well on our synthetic grid cell dataset but terminates with a higher error than both SIMPL and CEBRA, probably due to its misspecified emission model meaning it does not properly account for the Poisson nature of the spiking data. Furthermore, to give GPLVM the best chance and put it on level footing with CEBRA and SIMPL we initialised the its latent with behaviour. Note that in order to make use of all datapoint we had to restrict ourselves to using 1000 inducing points. Total compute time was just under 12 minutes.
> * Pi-VAE: Performed well, beating both CEBRA and GPLVM but finishing with an overall error (8.38 cm) about twice that of SIMPL and a compute time about 15.8x higher (we used a CPU for both). Like CEBRA, but less so, it's latent appears noisy which, also like CEBRA, we expect is due to the fact there is no explicit temporal dynamics/smoothing.
>
> We agree with the reviewer that these two methods are both more relevant that CEBRA which we initally compared to because of its popularity and ease of access. We will leave CEBRA there for completeness but we are happy to move it to the Appendix if the reviewer thinks this would help the clarity of the paper.
> Another viable contender could be a technique such as PfLDS (Gao, 2016). This satisfies the first three of our desiderata, but, to our knowledge, is not identifiable in the sense that there is no meaningful way to give behaviour as an input.
>
> Finally, related to benchmarking, we would also like to draw the reviewers attention to a new analysis and figure (Fig. 7), where we apply SIMPL to a hand-reaching macaque dataset from somatosensory cortex. This dataset on which SIMPL performs well, from the Neural Latents Benchmark suite, is commonly used in the LVM literature further supporting our claim that SIMPL is performant in comparison to similar LVM methods.
>
> We hope that our response and additional comparisons cleared the reviewer's concerns on this topic. We are happy to further clarify the position of SIMPL in the spectrum of data analysis methods by incorporating elements of this response in the Related Work section for a camera-ready version. We will be happy to address any remaining concerns the reviewer may have. Otherwise, we would be appreciate if the reviewer could adapt their score and recommend acceptance of our submission. Once again, we thank the reviewer for their time and insightful comments which, in our opinion, have led to substantial improvements in the manuscript.

---

> > ### Comment · Reviewer_gB66 · 2024-11-25
> >
> > I commend the authors on their thorough follow-up and additional evaluation. The response helps clarify how SIMPL relates to LVMs in neuroscience, but I unfortunately still believe that the current manuscript does not appropriately depict this large research space, nor does it clearly convey to a reader how the different features of SIMPL relate to the different features of the many latent variable models used in neuroscience.
> >
> > Specifically outlining the four desiderata is helpful in honing the conversation of relevant LVMs in neuroscience. Another relevant model might be VIND* -  a VAE type model with a dynamical latent space and Poisson observations. This might not meet the authors criteria for identifiable latent space, but there are indeed latent plots with nice scientific interpretation in their manuscript.
> > *Hernendez et al. Nonlinear Evolution via Spatially-Dependent Linear Dynamics for Electrophysiology and Calcium Data
> >
> > Note that Hurwitz et al and Gondur et al  - highlighted in the public comment - meet the four desiderata, but these models, like CEBRA, use behavioral data as an observation, and are not purely based on spiking like in SIMPL (save a behaviorally-informed initialization) and the others discussed.
> >
> > However, based on the current discussion and author clarifications, I don't believe VIND, nor these other deep LVMs are the best comparison, and I think the 4 desiderata are missing something crucial. It seems to me that an advantage to SIMPL is _interpretable_ tuning curves in addition to identifiable latent spaces. I agree with the authors that the GPLVM (w inducing points but with Gaussian emissions, or without inducing points but with Poisson emissions) is the closest class of models with very similar features to SIMPL. These approaches use identifiable smooth functions as the mapping from the latent space to observations, both achieved through kernel-based metrics. The authors have described both class of nonlinear models (those with NNs as well as those with GP or kernel-based functions) as having 'complex' tuning curves, as to separate them from linear approaches, but all of the models that use neural networks have the classic black-box problem and thus don't have this nice tuning curve interpretation (e.g. piVAE, CEBRA, VIND,  MMGPVAE (Gondur et al ), TNDM (Hurtwitz et al)). The authors make a point similar to this when they talk about identifiability.
> >
> > It seems to me that at it's core SIMPL is an alternative to the GPLVM - but even with the additional GPLVM comparison and evaluation on a simple synthetic example the manuscript doesn't make this clear. It is clearly favorable from a scalability point of view, which is nice, and having Poisson observations is another nice addition that could present some advantages to the Jensen et al GPLVM variant. However, the paper should demonstrate this and focuses specifically on the advantages and disadvantages of this approach to it's most closely related model(s) in neuroscience (that is, primarily the GPLVM- though a comparison to a linear LVM, say PfLDS or LFADs, and deep LVM, say PiVAE, could be helpful but in my opinion are not necessary).
> >
> > For example: the authors claim Poisson observations are a key desired feature of the model, and highlight it above as the important distinguishing feature from the Jensen GPLVM. If the addition of Poisson observations is indeed a core contribution above the competing approach, plots reinforcing the importance of this point would be crucial. Does SIMPL perform better than the GPLVM because of the Poisson observations, as the authors speculate? It might be nice to demonstrate this not just in a synthetic setting where Poisson observations are used to generate the data, but also in a real-world dataset. One could measure held-out spike prediction from the tuning curves comparing a Gaussian noise SIMPL to the GPLVM, to see if they match, and then are improved by adding Poisson observations to SIMPL, for example.
> >
> > Alternatively, the authors could focus on the scalability of their model while achieving similar performance in both tuning curve and latent identification to the (Poisson and Gaussian) GPLVM. They could further evaluate real-world latent-identification: e.g. does the latent space of SIMPL match position in place-cell or grid-cell data, or reaching position in motor data better than the GPLVM? And because scalability is a huge advantage to SIMPL, more thorough plots demonstrating would go a long way in strengthening the paper. E.g. in what limits is inference of SIMPL feasible compared to an inducing point GPLVM, and how does this speed-accuracy tradeoff change with amount of data and number of inducing points?
> >
> > In short, I believe this paper would greatly benefit from a re-write with these points in mind and a more thorough comparison to the GPLVM in a simulated and real setting.

---

> > > ### Author Response · Authors · 2024-11-25
> > > **Response (part 1)**
> > >
> > > ### Response summary (rewrite incoming)
> > >
> > > We thank the reviewer for their response and continued commitment to this paper. Their engagement has immensely helped position SIMPL within the landscape of neural-behavioural data analysis methods. We apologise for not having updated the introduction and related work section sooner; now that the reviewer has agreed that framing the background around our desiderata arguments clarifies the positioning of SIMPL **we will work immediately towards rewriting the relevant sections in a revised version of our manuscript in order that the reviewer will have time to assess before the end of the discussion period.** Please bear with us.
> > >
> > > **Regarding SIMPL vs. GPLVM** we will add a paragraph comparing GPLVM and SIMPL more precisely, as requested. However, we don't believe that GPLVM is so much closer to SIMPL than alternatives that the whole paper needs rewriting to exclusively frame SIMPL as an improvement on GPLVM. For instance, the GP prior on the latent dynamics significantly differs from the Markovian one of SIMPL. Under that specific light, models like PfLDS are closer to SIMPL than GPLVM (but they have other differences as previously discussed). SIMPL is a _new_ neural data analysis method, which makes its own modelling choices coming with their own trade-offs and should be considered as a new technique in its own right.
> > >
> > >
> > > On a related note, we thank the reviewer for proposing new ablations to understand the performance gains between SIMPL and GPLVM; we think running them would constitute interesting avenues for future work. However, in light of our last paragraph, we do not believe that these experiments are essential for (or would fit into the body of) a paper which does not frame itself as a direct variant of GPLVM. As a reminder, here are the major benefits of SIMPL as it stands:
> > >
> > > 1. We still maintain **SIMPL is the only technique satisfying all four desiderata** in a scalable way (see point-by-point below).
> > > 2. We demonstrate **SIMPL out-performs CEBRA, pi-VAE and, most importantly, GPLVM** on a non-trivial synthetic dataset. SIMPL returned tuning curves which matched ground truth much closer than the alternatives and had a latent error of less than half that of the next best competing model.
> > > 3. We show that **SIMPL is very fast**; we haven't identified any comparable technique that runs within even a tenth of SIMPL's speed.
> > > 4. We show **SIMPL already gives meaningful scientific insights** re. place cell analysis, Fig. 6. These finding are possible due to SIMPLs core features including its identifiability and scalability.
> > > 5. We show **SIMPL can be applied to spatial and motor-task datasets** (most existing techniques only show applicability to one domain e.g. Hurwitz 2021), beneficial to improve adoption of the technique across neuroscience subfields.
> > > 6.  **SIMPL is conceptually simpler than alternative techniques** enhancing its appeal to experimentalists (as pointed out by the reviewer).
> > >
> > > These are major, not incremental, improvements, with the potential to greatly enhance the efficiency of experimental analysis pipelines; we believe they are sufficient to justify the publication of our submission.
> > >
> > > We genuinely appreciate the reviewer's sincere dedication to advancing good science. As shown through the new experiments and benchmarks we've already conducted in response to the reviews, we are open to putting in substantial effort to improve the manuscript.

---

> > > ### Author Response · Authors · 2024-11-25
> > > **Response (part 2)**
> > >
> > > ### Point-by-point
> > > In the meantime, we respond to a few points individually:
> > >
> > > > **"Note that Hurwitz et al and Gondur et al - highlighted in the public comment - meet the four desiderata"**
> > >
> > > We checked and both models do not meet desideratum 1, which we consider "non-negotiable," as they both rely on exponential-linear-type tuning curves for their generative model.
> > >
> > >
> > > > **"neural networks have the classic black-box problem and thus don't have tuning curve interpretation (e.g. piVAE"**
> > >
> > > Just to clarify, most NN approaches like pi-VAE have a tuning curve component (in the sense they define a firing rate for all possible latent values $r_{it} = f_{\theta}(\mathbf{z}_t)_i$). We agree with the reviewer that this approach is less interpretable than the KDE approach taken by SIMPL.
> > >
> > > > **"Does SIMPL perform better than the GPLVM because of the Poisson observations"**
> > >
> > > It possible that this is why. It is also possible that GPLVM suffers from innaccuracies due to using sparse induction points. Or it has to do with the intricacies of the inference procedures both methods employ. We are open to performing the Poisson analysis yet hesitant because this would require a substantial rewrite of SIMPL codebase (current likelihood functions have the Poisson-ness baked in). From a performance-only perspective it is also a moot point: SIMPL _does_ perform better as shown in Fig. 5 and we would be happy to see future work on figuring out why. If the reviewer would like we can sweep the number of induction points (up to our own compute limits) to see if GPLVM benefits.
> > >
> > > > **"does the latent space of SIMPL match position in place-cell or grid-cell data, or reaching position in motor data better than the GPLVM?""**
> > >
> > > We are open to applying GPLVM to the hippocampal dataset and including these results in the camera-ready. However, it is not immediately clear how such a comparison would inform which of the techniques is "better". Evaluating an LVM by how similar it's latent is to behaviour is objectively _incorrect_ (though it can be scientifically revealing for other reasons) as it makes the assumption that behaviour == the latent. For this reason we focus on our sythetic dataset where ground truth is known and thus comparisons can be made concrete using the L2-error between latent and ground truth. Although previous studies, such as Fig. 3 in the P-GPLVM paper, have used correlation with behaviour as their benchmark metric, we find this approach problematic. Nonetheless, it could still be interesting the check if GPLVM finds the same changes in the place fields as SIMPL.

---

> > > ### Author Response · Authors · 2024-11-28
> > > **Response: Sections rewritten and additional benchmark/comparisons to GPLVM added**
> > >
> > > We have just uploaded a revised version of our manuscript.
> > > In response to the reviewers comments we have adjusted the paper to (i) more tightly constrain the scope of SIMPL and (ii) more clearly describe, and compare to, its most relevant alternatives. We elaborate on these changes below:
> > >
> > >
> > > 1. We have **completely rewritten the Related Work** section to more comprehensively cite existing literature and frame SIMPL within the field of LVMs, clarifying its benefits with respect to alternatives. We have organised this section around the desiderata described above. We then give more focussed discussions on CEBRA, pi-VAE, GPLVM (+ associated models); three good candidates for comparison to SIMPL in that they do not place restrictive linear assumptions on the tuning curves and can naturally exploit behaviour.
> > >
> > >     _All_ the citations mentioned in this review forum have now been added to the manuscript.
> > > 2. **Made edits to the Introduction, Discussion and Methods** with the intention of more clearly defining, indeed _shrinking_, the scope of SIMPL. The intention here is to make clear that existing techniques were developed in different contexts with different goals in mind (some, for example, were developed entirely outside of neuroscience). Thus SIMPL is not a be-all and end-all solution to LVMs but a specific one. In the Methods we added a sentence clarifying the benefits of our KDE M-step over neural network approaches in terms of interpretability.
> > > 3. **Added a table to the appendix summarising _all_ the studies we have discussed** as well as some others. This table, containing 24 methods, clarifies each methods position with respect to our desiderata. We hope that this goes some way towards, as the reviewer suggested, "appropriately depicting this large research space, and conveying to a reader how the features of SIMPL relate to the different features of the many latent variable models used in neuroscience". We also hope this will help future readers comprehend the field of LVMs for neural data analysis and drive further development.
> > > 4. **Additional comparisons to GPLVM-style methods**. Following the reviewers suggestions to better compare SIMPL to GPLVM-based techniques we have added an additional benchmark against GPDM (Wang, 2005), a variant of GPLV which imposes smooth latent dynamics through a Gaussian process prior. GPDM does not come with an induction point variant thus we were restricted to running it on a subset (10,000 / 36,000) of data points. We found its performance to be comparable to GPLVM.
> > >
> > >     Furthermore, following the suggestions of the reviewer, we investigated the impact of GPLVMs misspecified noise model by running it on a control dataset generated with the same grid cell tuning curves but instead of a Poisson noise model, a Gaussian noise model. For the Gaussian data, like with out original Poisson data, the tuning curves specified the means of the observations and we set the standard deviation to a fixed value of 0.1. The results are shown in a figure in a new figure added to the Appendix. While, GPLVM performed slightly better on the control dataset (as expected) the improvement remained small compared to the difference between GPLVM and SIMPL. Other differences between SIMPL and GPLVM (e.g. the inducing point approximation used for GPLVM, optimisation issues, or other forms of misspecification/model differences) must be more important.
> > >
> > >
> > > Once again we are very grateful for the reviewers continued engagement and insightful comments which have led to meaningful improvements in our manuscript. We have tried our best to handle all of their concerns an will remain available for any further discussion or clarification which might be required.

---

> > > > ### Author Response · Authors · 2024-12-01
> > > > **Follow-up**
> > > >
> > > > As the deadline for responding is approaching we would like to ask whether the reviewer has had time to consider our response and the changes we made to the manuscript. We remain available for discussion if needed.

---

> ### Author Response · Authors · 2024-11-25
> **Follow-up**
>
> As the discussion period is coming to a close we would like to ask whether the reviewer has had the time to consider our comments and, in light of these, reconsider their score. To summarise; the reviewer's major concern was that we had not benchmarked SIMPL against sufficient nor relevant existing methods. Largely we were in agreement with the reviewer and, as such, responded with:
>
> * **Two new benchmarks** against more relevant LVM techniques mentioned by the reviewer: pi-VAE and GPLVM-with-inducing-point.
>      * These techniques work well on our dataset but still significantly underperform SIMPL in terms of both absolute final error and compute time.
>      * These have been added to Fig. 5.
>      * Both our new benchmarks (assuming a fixed number of inducing points for GPLVM) have linear time complexity, matching SIMPL. But our experiments suggest they are both slower by approximately a factor of 10x.
> * A thorough **review of the literature** flagged by the reviewer
> * Substantial **discussion on what methods constitute relevant benchmarks** and why.
>
> Full details can be found in our original response above. We thank the reviewer again for their suggestions and ask if they would let us know if they have any remaining major concerns.

---

### Official Review · Reviewer_Kcef · 2024-11-04

**Soundness:** 3
**Presentation:** 3
**Contribution:** 3
**Rating:** 8
**Confidence:** 3

**Summary:**

The paper introduces a new method they call SIMPL, an EM-style algorithm, aiming to recover low-dimensional, time-evolving latent variables from high-dimensional neural activity in large neural datasets. Their  approach fits tuning curves to observed behaviour and iteratively refines these through a two-step process (EM like). The originality of SIMPL lies in its novel combination of expectation-maximization (EM) techniques with behavior-driven initialization, providing a straightforward yet effective approach to refine neural representations. It’s main advantage is that it is a scalable and fast approach compared to existing popular methods like CEBRA.

**Strengths:**

SIMPL has only two hyperparameters. Minimal hyperparameters make it practical for experimentalists.

Rigorous comparison with existing methods (CEBRA).It outperforms CEBRA (whose latent embedding was noisier and had larger final error) and  is over 30 faster.

The capability to achieve results over 30 times faster than comparable methods while maintaining accuracy makes SIMPL particularly attractive for large-scale neuroscience research

Careful experiments on synthetic as well as real data.

**Weaknesses:**

Limited Evaluation on Non-Spatial Tasks: While the paper demonstrates impressive results in the domain of spatial navigation, it would benefit from evaluation on other tasks, such as the ones tested in the CEBRA paper so that the two methods can be directly compared in more diverse settings.

Scalability Concerns for High-Dimensional Latents: The authors mention that kernel density estimation (KDE) may not scale well to high-dimensional latent spaces. This potential limitation needs more elaboration. What alternatives might be suitable when dealing with truly high-dimensional data.

**Questions:**

How does the method perform with smaller neural populations? Can SIMPL perform well when there are limited neurons recorded. Some downsampling analysis on neural data can be helpful to check if the method is sensitive to neuron number.

How much data is required to optimize SIMPL? How does the mother deal with short recordings? For example, would a short half hour recording be sufficient to fit the model?

The author mentioned that they trained CEBRA on the synthetic grid cell data using out-of-the-box hyperparameters training for the default 10000 iterations. How much of the performance inferiority of CEBRA VS SIMPL comes from not carefully searching through the hyperparameter space? Is the comparison fair? Would increasing the iterations improve the performance of CEBRA?


The author mentioned that SIMPL finds a modified latent space with smaller, more numerous, and more
uniformly-sized place fields, suggesting the brain may encode space with greater resolution than previously thought. How could the author disentangle the possibility that this observation is a result of artifacts by their method versus reflecting true representation feature of the hippocampus?

---

> ### Author Response · Authors · 2024-11-20
> **Response: New analysis on a non-spatial dataset and sweep testing data-size requirements**
>
> We thank the reviewer for their time take to thoroughly assess our paper. In response we have run new analyses as well has hyperparameter sweeps ot the manuscript. As a result we believe the paper is now in a much stronger position to be accepted. Our point-by-point response is as folllows:
>
> > **"SIMPL has only two hyperparameters. Minimal hyperparameters make it practical for experimentalists."**
>
> Furthermore, we have now added a new figure to the Appendix (Fig. 8) where we sweep across these two parameters and show the model performs well across a large area of parameter space.
>
> > **"Rigorous comparison with existing methods (CEBRA). It outperforms CEBRA (whose latent embedding was noisier and had larger final error) and is over 30 faster."**
>
> We have also added a comparison to GPLVM (and equivalent but Gaussian process based technique) and Pi-VAE as suggested by reviewer gB66.
>
> > **"Limited Evaluation on Non-Spatial Tasks...[it] would benefit from evaluation on other tasks, such as the ones tested in the CEBRA paper"**
>
> Following this review we have now tested SIMPL on one of the Neural Latents Benchmark datasets (specifically the [Area2_Bump data](https://neurallatents.github.io/datasets) from somatosensory cortex for a macaque doing a centre-out reaching task collected by Chowdhury and Miller), the same dataset as used in the CEBRA paper. We find SIMPL works well (see the revised manuscript for a more detailed discussion of the results). In the results we have now added a new section and figure (Fig. 7) detailing our findings briefly summarised as follows:
> We tested 3 versions of SIMPL on the hand-reaching data.
> - SIMPL2D(position): The latent is initialised with the monkeys x- and y-hand position.
> - SIMPL2D(velocity): The latent is initialised with the monekys $v_x$- and $v_y$-hand velocity.
> - SIMPL4D(postion&velocity): A 4D latent is initialised with $x$, $y$, $v_x$ and $v_y$.
> In all three models SIMPL optimises the latent variable (the test-log-likelihood improves), uncovering a smooth latent variable correlated to (but substantially different from) the behavioural initiates. Corresponding tuning curves revealed neurons with "hand-position-like" or "hand-velocity-like" selective receptive fields. The 4D version of SIMPL performed better than either 2D version, revealing disentangled latents with a higher overall log-likelihood than either of the 2D models. Our finding reveal:
> 1. SIMPL can be applied to the types of non-hippocampal datasets commonly used in the LVM literature.
> 2. Both position and velocity _seperately_ do a good job explaining the latent dynamic but position and velocity _combined_ is better.
> 3. In the optimised latent space neurons have localised receptive fields reminiscent of place cells in the hippocampus.
> 4. SIMPL can be extended beyond the 2-dimensional latent spaces initially tested.
>
> > **"Scalability Concerns for High-Dimensional Latents...What alternatives might be suitable when dealing with truly high-dimensional data?"**
>
> It is true that we remain cautious about SIMPL's performance in very high-dimensional latent spaces. This is primarily because, as a function approximation technique, KDE suffers from the "curse of dimensionality". We believe that neural-network based approaches might have the potential to perform better in such high-dimensional scenarios, a point we made in the orignal submission but will now further clarify. However, these models are harder to optimize, often requiring tedious hyperparameter tuning, and may put off prospective scientists.
>
> > **"How does the method perform with smaller neural populations?...How much data is required to optimize SIMPL?"**
>
> We thank the reviewer for bringing this important point to our attention. These are important questions. We repeated our synthetic grid cell experiment using decreasing amounts of data (fewer neurons and shorter duration), the results are shown in a new subpanel of Fig. 3 (panel e). In summary we find that the size of our original dataset (225 neurons, 60 minutes) was much larger than actually required. SIMPL still performs well (and runs _much_ faster) with only 50 neurons and 10 minutes of data. Of the two, we found that number of neurons is more important than the duration, with performance dropping off sharply when $\leq$ 20 neurons are used. This is easily achieved by most modern neural datasets and performance of SIMPL in lower data regimes is empirically confirmed in our new analysis on a somatosensory dataset which has only 65 neurons and 37 minutes.

---

> ### Author Response · Authors · 2024-11-20
> **...(response continued)**
>
> > **"How much of the performance inferiority of CEBRA VS SIMPL comes from not carefully searching through the hyperparameter space? Is the comparison fair? Would increasing the iterations improve the performance of CEBRA?"**
>
> We don't think so. The relative underperformance of CEBRA is not because it hasn't converged but rather because it does not assume any dynamics on the underlying latent variable which would "smooth" (or "denoise") it. We made this comment in the original submission but will make it clearer in the camera-ready version. In CEBRA each spike bin is treated independently and is thus subject to its own irreducible noise. Conversely, SIMPL (and other dynamical LVMs) assumes the latent follows a linear dynamical systems thus employs Kalman smoothing which denoises the latent substantially.
>
> > **"How could the author disentangle the possibility that this observation is a result of artifacts by their method versus reflecting true representation feature of the hippocampus?"**
>
> This is a an important question which we spent time addressing in the original submission (see paragraph 4 of the section on hippocampal dataset). In summary, we ran SIMPL on a control dataset of spikes sampled from behaviour and behaviour-fitted tuning curves. Results are shown in a grey shade on Fig. 6. For these spikes behaviour and behaviour-fitted tuning curves should be stable (since they are, by definition, exactly the generative model) and any changes reflect artifacts of the SIMPL algorithm. We observe no significant changes to the place fields for the control spikes suggesting the changes observed in the real neural data are not artifacts
>
> Lastly, we thank the reviewer again for their thorough review and would appreciate if they would reconsider their score in light of our new additions to the manuscript which, we believe, have strengthened it substantially.

---

> ### Author Response · Authors · 2024-11-25
> **Follow-up**
>
> As the discussion period is coming to a close we would like to ask whether the reviewer has had the time to consider our comments and, in light of these, reconsider their score. To summarise; along with a point-by-point discussion to each of the reviewers questions we believe we have handled their most important concerns by making the following changes to the paper:
>
> * Adding a new analysis/figure where we **analyse a non-spatial motor-task** dataset.
> * Adding a new subpanel to fFig. 3 where we **test performance against dataset size**, and show SIMPL works well with significantly smaller datasets.
>
> Full details can be found in our original response above. Please would the reviewer let us know if they have any remaining major concerns.

---

> > ### Comment · Reviewer_Kcef · 2024-11-27
> >
> > Thank you to the authors for their detailed response and additional experiments. I have raised my score in light of the new experiments.

---

### Official Review · Reviewer_6vN3 · 2024-11-09

**Soundness:** 3
**Presentation:** 3
**Contribution:** 3
**Rating:** 6
**Confidence:** 3

**Summary:**

The paper introduces SIMPL, an EM-style algorithm for refining latent variables and tuning curves from spiking neural data. The key idea is using behavior as initialization and combining kernel density estimation with Kalman smoothing in an iterative optimization framework. The authors validate their approach on synthetic datasets and real hippocampal recordings, showing improvements in both computational efficiency and biological interpretability compared to CEBRA (a recent popular deep learning approach for learning latent embeddings from neural and behavioural data)

**Strengths:**

- The paper is a fresh departure from current latent variable models relying on deep neural networks, with a number of practical advantages (e.g., fast optimization, minimal hyperparameters (only 2), simple implementation based on standard components (Kernel Density Estimation and Kalman smoothing))
- The paper reveals scientific insights through the use of the proposed method (finer structure in place field representations, new interpretation of place field size distribution, relationship between behavioral uncertainty and neural encoding)
- The authors do a good job presenting the approach clearly and acknowledging the limitations of the proposed method.

**Weaknesses:**

The proposed approach relies on strong assumptions that seem to limit the application of the method for more complex cases beyond the datasets used here (e.g., settings where behavioral recordings are missing, higher dimensional latent states, latent spaces not dominated by behavior, non-smooth latent trajectories). It would be great to demonstrate the applicability of the approach on commonly used datasets in the current literature on LVMs in neuroscience (e.g., Neural Latents Benchmark).

**Questions:**

- The authors mention improved identifiability of the proposed approach. Is that solely because of the behavioral initialization?
- The authors say: "we see SIMPL as a specific instance of a broader class of latent optimization algorithms, ..." and then move on to the idea of replacing KDE with neural networks. Can the authors elaborate on that?
- Can the authors provide the intuition for choosing the kernel bandwidth in the KDE step? This seems critical for the method's performance.

---

> ### Author Response · Authors · 2024-11-20
> **Response: New analysis on new neural dataset and parameter sweeps**
>
> We thank the reviewer for their time taken to assess our paper. We were especially happy to read, and would like to reiterate, their point about SIMPL being a "fresh departure" from previous techniques. We have made a number of changes to the manuscript in response to their review, several new figures and sub-panels as well as an entirely new analysis on a non-hippocampal dataset from the Neural Latents Benchmarks suite. We hope that these changes address their concerns.
>
> > **"Higher dimensional latent states...and datasets in the current literature on LVMs"**
>
> Following this review we have now tested SIMPL on one of the Neural Latents Benchmark datasets (specifically the [Area2_Bump data](https://neurallatents.github.io/datasets) from somatosensory cortex for a macaque doing a centre-out reaching task collected by Chowdhury and Miller), the same dataset as used in the CEBRA paper. We find SIMPL works well (see the revised manuscript for a more detailed discussion of the results). In the results we have now added a new section and figure (Fig. 7) detailing our findings briefly summarised as follows:
> We tested 3 versions of SIMPL on the hand-reaching data.
> - SIMPL2D(position): The latent is initialised with the monkeys x- and y-hand position.
> - SIMPL2D(velocity): The latent is initialised with the monekys $v_x$- and $v_y$-hand velocity.
> - SIMPL4D(postion&velocity): A 4D latent is initialised with $x$, $y$, $v_x$ and $v_y$.
>
> In all three models SIMPL optimises the latent variable (the test-log-likelihood improves), uncovering a smooth latent variable correlated to (but substantially different from) the behavioural initiates. Corresponding tuning curves revealed neurons with "hand-position-like" or "hand-velocity-like" selective receptive fields. The 4D version of SIMPL performed better than either 2D version, revealing disentangled latents with a higher overall log-likelihood than either of the 2D models. Our finding reveal:
> 1. SIMPL can be applied to the types of non-hippocampal datasets commonly used in the LVM literature.
> 2. Both position and velocity _seperately_ do a good job explaining the latent dynamic but position and velocity _combined_ is better.
> 3. In the optimised latent space neurons have localised receptive fields reminiscent of place cells in the hippocampus.
> 4. SIMPL can be extended beyond the 2-dimensional latent spaces initially tested.
>
> > **"Settings where the behavioural recordings are missing or latent spaces not dominated by behaviour"**
>
> We have already shown in Fig. 4 that SIMPL still works well _without_ behaviour. The "catch", in such instances, is that you lose the identifiability guarantees that come when a relevant behavioural variable is used for initialisation. Because of this we acknowledge and agree with the reviewer that the most likely use-case for SIMPL will be to discover/refine tuning curves in neural data which is well characterized by a behavioural variable already available to the user.
>
> Although this is, on the face of it, a limitation, it reflects a very conscious choice made to bypass a fundamental issue in latent variable modelling. Latent variable modelling is fundamentally hard and an entirely general purpose technique which is (i) totally assumption free and (ii) computationally cheap and hassle-free across all datasets simply doesn't (perhaps never will) exist. We contest that many popular techniques have actually _under relied_ on behaviour which has resulted in methods which are overly complex (putting off non-theoreticians) or compute heavy. Our observation is that in a majority of neuroscience experiments the latent space _is_ closely related to a behaviour which is (or could be easily) measured concurrently - in such a majority of cases SIMPL strikes a good balance between simplicity, interpretability and computational efficiency. We will add a discussion of this point in the revised manuscript.

---

> ### Author Response · Authors · 2024-11-20
> **...(response continued)**
>
> >**"Non-smooth latent trajectories"**
>
> We agree with the reviewer that it is important to test SIMPL when the latent variable is non-smooth. Currently the amount of smoothness in the latent trajectory is controllable via the velocity hyperparameter and can be set arbitrarily close to zero (at which point the decoding procedure amounts to pure maximum likelihood estimation, capable of modelling non-smooth latents).
>
> To further alleviate concerns over whether SIMPL can model non-smooth latents, we generated a synthetic "replay" dataset. In this dataset the latent and behaviour are identical except for regular, brief instances where the latent siccontinuously jumps to a new location in the environment then jumps back. With the same parameters as before SIMPL is able to accurately recover the latent, correctly recapitulating the discontinuous jumps. In summary, although SIMPL is biased towards smooth latents this is just a prior and, with reasonable parameter settings, non-continuous latents can be modelled as well. These new results are summarised in a new figure (Fig. 9) in the appendix.
>
> > **"The authors mention improved identifiability of the proposed approach. Is that solely because of the behavioral initialization?"**
>
> Yes, we find that the initialization near behaviour results in convergence on a latent space which is the same as the ground truth (at least, in our sythetic experiments, which are the only examples where true ground truth is knowable). Results in favour of this hypothesis include that fact that when behavioural intialisation is removed SIMPL learns a warped/fragmented latent space. As well as the fact that CEBRA, which was not initialised with behaviour, did not find such a good latent and it's grid fields appear slightly warped relative to the ground truth.
>
> > **"The authors say: "we see SIMPL as a specific instance of a broader class of latent optimization algorithms, ..." and then move on to the idea of replacing KDE with neural networks. Can the authors elaborate on that?"**
>
> This point makes clear that there could, in theory, be many ways of fitting tuning curves. We have chosen KDE because it is simple, interpretable and fast but a neural network (e.g. trained to output firing rates given a latent) could be used instead and would come with its own advantages and disadvantages which we have not studied here. We have not tested this but it is an interesting avenue for future work. We will rewrite this section to make this clearer.
>
> > **"Can the authors provide the intuition for choosing the kernel bandwidth in the KDE step? This seems critical for the method's performance."**
>
> There are some general heuristics for choosing the kernel bandwidth, for exampled see Silverman's rule of thumb.  Roughly, the goal is to choose the smallest bandwidth allowed by the data without over- or under-fitting: too small and individual spikes will be resolved, too big and high-frequency structure in the receptive fields will be smoothed. In practice SIMPL is not particulary sensitive to this parameter as shown in our new hyperparameter sweep performed in respjse to the reviewers concerns (see appendix Fig. 8) where we find that performance is good across an order of magnitude of kernel bandwidths (between 0.5 cm and 5 cm) and an order of magnitude of speed priors (between 0.1 ms$^{-1}$ and 1 ms$^{-1}$). These ranges will, of course, be dataset dependent, but they do suggest that the range of appropriate hyperparameters might not be very sharp. In summary, our new hyperparameter sweep shows SIMPL has only soft dependence on the hyperparameter.
>
> On the basis of our adjustments and additional analysis on the Neural Latents Benchmark dataset, we believe that the paper is now in a much stronger position to be accepted. Please would the reviewer let us know if they have any further concerns or questions will might prevent them recommending our paper for acceptance.

---

> ### Author Response · Authors · 2024-11-25
> **Follow-up**
>
> As the discussion period is coming to a close we would like to ask if the reviewer has had the time to consider our comments and, in light of these, reconsider their score. To summarise; along with a point-by-point discussion we believe we have handled their most important concerns by:
>
> * Adding a new analysis/figure where we **analyse a non-spatial motor-task** dataset.
> * Adding a new experiment/figure where we show **SIMPL can account for discontinuous latent trajectories**.
> * Performing a **sweep across the kernel bandwidth and velocity prior hyperparameters** showing SIMPL performance is _not_ sharply dependent on their values.
>
> Full details can be found in our original response above. Please would the reviewer let us know if they have any remaining major concerns.

---

> > ### Comment · Reviewer_6vN3 · 2024-11-26
> >
> > I thank the authors for their detailed response to me and the other reviewers.
> > I belive the additional experiments and the updated writing better highlight the position of the proposed approach within the landscape of lvms in neuroscinece. While the potential impact of the method remains a concern, I think the paper and the is constructive addition to the literature of neural-behavioural data analysis. I have raised my score.

---

### Public Comment · ~Cole_Lincoln_Hurwitz1 · 2024-11-13
**Missing citations for neural-behavioral modeling**

There are a bunch of methods for neural-behavioral modeling which are not cited in this paper. It would be good to include these references as building latent variable models that exploit behavior is a very active field of study.

- Sani, Omid G., et al. 2021: This work proposes a linear dynamics approach for modeling neural-behavioral data.

- Hurwitz et al. 2021: This work proposes a sequential VAE for modeling neural-behavioral data.

- Gondur et al. 2024: This work proposes a multi-modal gaussian process variational autoencoder for neural-behavioral data

- Sani et al. 2024: This work proposes an RNN-based architecture for neural-behavioral data.

Rabia Gondur, Usama Bin Sikandar, Evan Schaffer, Mikio Christian Aoi, and Stephen L Keeley. Multi-modal gaussian process variational autoencoders for neural and behavioral data. In International Conference on Learning Representations, 2024.

Sani, Omid G., et al. "Modeling behaviorally relevant neural dynamics enabled by preferential subspace identification." Nature Neuroscience 24.1 (2021): 140-149.

Cole Hurwitz, Akash Srivastava, Kai Xu, Justin Jude, Matthew Perich, Lee Miller, and Matthias Hennig. Targeted neural dynamical modeling. Advances in Neural Information Processing Systems, 34:29379–29392, 2021.

Omid G Sani, Hamidreza Abbaspourazad, Yan T Wong, Bijan Pesaran, and Maryam M Shanechi. Modeling behaviorally relevant neural dynamics enabled by preferential subspace identification. Nature Neuroscience, 24(1):140–149, 2021.

---

> ### Author Response · Authors · 2024-11-20
> **Thanks**
>
> We'll take a look at these references and filter them into the manuscript for the camera-ready version.

---

### Author Response · Authors · 2024-11-20
**Summary of the reviews and our response**

_Nb: PLEASE RE-READ this has been edited to reflect further discussion with reviewer gB66_

Based on the reviews and subsequent follow-up discussion, we have submitted a revised manuscript which has undergone significant improvements.

**To summarise _strengths_**, the reviewers appreciated the novelty of our paper stating that it is "a fresh departure from current latent variable models relying on deep neural networks" (6vN3) as well as its computation efficiency and scalability (Kcef, 5Qja, gB66) and that "the simplicity of the approach has potentially some appeal to experimentalists who want to avoid more complex latent variable models" (gB66). They noted that it reveals new scientific insights (6vN3) and that we made "careful experiments on synthetic as well as real datasets"(Kcef, gB66) as well as "rigorous comparisons with existing methods"(Kcef) . They also commented that the paper was well presented (5Qja, 6vN3), the figures were clear (5Qja) and that we "acknowledged the limitation of the proposed method" well (6vN3)

**To summarise _weaknesses_**, there were two primary concerns. Firstly, three of the reviewers (5Qja, 6vN3, Kcef) wanted to see SIMPL tested on additional non-spatial datasets, all pointing towards the same Macaque hand-reaching dataset used in previous LVM studies (we have now done this, consequently the reviewers have raised their scores to accept/strong accept-status). Secondly, reviewer gB66's primary concern was that we have not made sufficient, nor relevant comparisons to alternative techniques (we have discussed this in depth, added three more methods to our benchmarks, rewritten the related work and added a technique-comparison table to the appendix). The reviewers also had other more minor concerns about whether SIMPL performance depends strongly on the choice of hyperparameters (6vN3, we tested this and it did not), dataset size (Kcef, likewise), or continuity of the latent variable (6vN3, likewise). To a large extent, we are in agreement with the reviewers regarding all of these concerns. Consequently we have run new benchmarks, performed new experiments/hyperparameter sweeps, run SIMPL on the Macaque dataset and rewritten the related work section, all of which leave the manuscript in a much stronger position. Our most substantial changes are summarised in the following table:

**Summary of most notable changes**

| Addition | Comment | Relevant reviewers |
|-------|-------|------|
| **Macaque dataset analysis** |SIMPL was applied to a somatosensory macaque dataset and performed well demonstrating its applicability to non-spatial and higher-dimensional latent spaces. | 6vN3, Kcef, 5Qja |
| **Additional benchmarks**| We have now benchmarked SIMPL against three additional, more relevant methods (pi-VAE, GPLVM and GPDM as well as the original CEBRA). In all cases SIMPL performs best and computes fastest. These comparisons cover the range of relevant LVM features (see extended discussion below), confirming SIMPL's superiority. | gB66, 5Qja |
| **Hyperparameter sweep** | A new figure has been adding confirming the robustness of SIMPL to its two main hyper parameters ($v$ and $\sigma$). | 6vN3 |
| **Dataset size sweep** | We ran SIMPL on datasets of increasing smaller and smaller size (both neuron count and duration) to see how it performs in the low-data limit. | KceF |
| **Discontinuous latent experiment** | We ran a new experiment where the latent is non-continuous and confirmed SIMPL can accurately recapitulate it (despite having an explicit smoothness prior) | 6vN3 |
| **Discussion of relevant alternatives** | Additional discussion of what models constitute relevant alternatives and why. | gB66 |
| **Rewritten Related Work** | To better explain the field of LVMs and frame SIMPL in relation to alternative techniques specifically clarifying a minimal set of unique features SIMPL and only SIMPL satisfies. | gB66 |
| **Added missing citations** | All citations mentioned on this forum have not been added to the manuscript | gB66, C. Hurwitz (public commenter) |

All new figures have been added to a revised manuscript in order that the reviewers can assess them. In this manuscript new text is shown in blue.

---

### Meta-Review · Area_Chair_PjVJ · 2024-12-23

**Metareview:**

This paper introduces a latent variable model (LVM) for neural and behavioural representations. The model is able to learn low-dimensional latent variables from high-dimensional neural activity. The learning algorithm is based on the Expectation-Maximization framework and it is more scalable than competitive LVMs and can be applied to large datasets.

The reviewers appreciate that this method is simple (only two hyperparameters), yet intuitive and effective. In particular, scalability gains are high, with results showing a 30x boost over comparable methods without losing accuracy. Furthermore, the reviewers find it interesting that the paper reveals scientific insights, such as the finer structure in place field representations and the relationship between behavioral uncertainty and neural encoding.

There has been extensive discussion about adding clarifications, and the revised version has largely addressed those. With the addition of new experiments during the rebuttal period, the overall experimental evaluation section is stronger.

However, there is one concern remaining: the comparisons are done in a simulated setting (figure 4) rather than showing the comparison in the real-world setting. This doesn’t seem to be a major concern about the convincingness of the method per se, but it is rather a concern about whether these evaluations support claims like _“when it is applied to a large rodent hippocampal dataset, SIMPL efficiently finds a modified latent space with smaller, more numerous, …”_. The wording of these claims should be easy to modify appropriately for the camera-ready version.

Overall, I recommend the paper for acceptance.

**Additional Comments On Reviewer Discussion:**

There has been satisfactory discussion where all reviewers and authors were engaged. A lot of the discussion focused on clarifications and suggestions for improving the text, in particular reviewer `5Qja` gave many useful suggestions which the authors eventually incorporated in the manuscript.

Besides, most of the discussion focused on experimental evaluation. The reviewers requested comparison on non-spatial datasets, and the authors have presented new experiments which were appreciated by the reviewers. The authors also offered new ablation studies, as requested by reviewer `6vN3` (for hyperparameters) and `Kcef` (for dataset  size).

There has also been discussion about placement of this work within the literature and generally how impactful the method is expected to be in the real-world, given the evaluation section. On this aspect, the reviewers partially align, and in private discussions it is acknowledged that evaluation of more baselines on the real setting would help, however not all reviewers judge this omission equally in terms of importance.

Overall, even if there is not full consensus, it seems that the reviewers are _generally_ positive about this paper.

---

### Decision · Program_Chairs · 2025-01-22

Accept (Poster)